# DeCap: Decoding CLIP Latents for Zero-Shot Captioning via Text-Only Training

**Wei Li**[1]    **Linchao Zhu**[1]    **Longyin Wen**[2]    **Yi Yang**[1*]
[1]CCAI, Zhejiang University    [2]ByteDance Inc., San Jose, USA
{weili6,zhulinchao,yangyics}@zju.edu.cn   longyin.wen@bytedance.com

## Abstract

Large-scale pre-trained multi-modal models (e.g., CLIP) demonstrate strong zero-shot transfer capability in many discriminative tasks, e.g., image classification. Their adaptation to zero-shot image-conditioned text generation tasks has drawn increasing interest. Prior arts approach to zero-shot captioning by either utilizing the existing large language models (e.g., GPT-2) or pre-training the encoder-decoder network in an end-to-end manner. However, the large language models may not generate sensible descriptions due to the task discrepancy between captioning and language modeling, while the end-to-end pre-training requires paired data and extensive computational resources. In this work, we propose a simple framework, named DeCap, for zero-shot captioning. We introduce a lightweight visual-aware language decoder. This decoder is both data-efficient and computation-efficient: 1) it only requires the *text* data for training, easing the burden on the collection of paired data. 2) it does not require end-to-end training. When trained with text-only data, the decoder takes the text embedding extracted from the off-the-shelf CLIP encoder as a prefix embedding. The challenge is that the decoder is trained on the text corpus but at the inference stage, it needs to generate captions based on visual inputs. Though the CLIP text embedding and the visual embedding are correlated, the *modality gap* issue is widely observed in multi-modal contrastive models that prevents us from directly taking the visual embedding as the prefix embedding. We propose a training-free mechanism to reduce the modality gap. We project the visual embedding into the CLIP text embedding space, while the projected embedding retains the information of the visual input. Taking the projected embedding as the prefix embedding, the decoder generates high-quality descriptions that match the visual input. The experiments show that DeCap outperforms other zero-shot captioning methods and unpaired captioning methods by a large margin on the typical image captioning benchmarks, i.e., MSCOCO and NoCaps. We apply DeCap to video captioning and achieve state-of-the-art zero-shot performance on MSR-VTT and ActivityNet-Captions. The code is available at `https://github.com/dhg-wei/DeCap`.

## 1 Introduction

The goal of image captioning is to automatically generate descriptions for given images. Models (Anderson et al., 2018; Lu et al., 2017; Rennie et al., 2017; Zhang et al., 2021; Huang et al., 2021) trained on human-annotated image-text pairs have achieved impressive results on typical image captioning benchmarks. However, due to the small size and limited visual concepts of human-annotated datasets, these models generalize poorly to images in the wild (Agrawal et al., 2019; Tran et al., 2016; Wu et al., 2018). In this paper, to reduce the reliance on human-annotated paired data and improve the generalization in real-world captioning scenarios, we propose a new zero-shot captioning framework that requires text-only data for training.

Pre-training on web-scale noisy paired data has been demonstrated to be effective in learning robust multi-modal representations (Radford et al., 2021; Jia et al., 2021; Li et al., 2021; Alayrac et al., 2022; Yu et al., 2022a; Wang et al., 2022; Zhu & Yang, 2020). Changpinyo et al. (2021) and

---

*Yi Yang is the corresponding author.

Wang et al. (2021b) use web-scale image-text pairs to train a captioning model and achieve great improvements on MSCOCO (Chen et al., 2015) and NoCaps (Agrawal et al., 2019) through the pretraining-finetuning paradigm. However, these models show inferior zero-shot captioning performance on MSCOCO, indicating that these methods still rely on human-annotated paired data for fine-tuning. Besides, training with the captioning objective on web-scale data is not efficient, e.g., Wang et al. (2021b) train their model on ALIGN (Jia et al., 2021) and C4 (Raffel et al., 2020) about 1M steps using 512 TPU v3 chips (Jouppi et al., 2017).

Instead of directly training a captioning model in an end-to-end manner on web-scale image-text pairs, another line of work (Tewel et al., 2022b; Su et al., 2022) achieves zero-shot captioning by combining existing pre-trained models. Specifically, they use a pre-trained multi-modal model CLIP (Radford et al., 2021) to guide a pre-trained language model (PLM), i.e., GPT-2 (Radford et al., 2019), to generate sentences that match the given image. However, the inference speed of these methods is slow because each word generation involves a CLIP text encoder forward. Besides, language models pre-trained on various documents from webpages do not match well with captioning tasks that aim to describe visual concepts and their relationships in a given image, resulting in inferior performance on image captioning benchmarks.

In this paper, we propose a new framework, named DeCap, for zero-shot captioning. We aim to decode sensible visual descriptions from the CLIP multi-modal embedding space. We do not use paired image-text data during the decoder pre-training but only leverage the text data. This is more flexible and efficient when the alignment between images and texts became noisier. Our DeCap framework is described below: During **pre-training**, the text decoder is trained from scratch. The goal is to invert the CLIP text encoder, i.e., a sentence is first encoded into an embedding by the CLIP text encoder and later reconstructed by our text decoder. The decoder takes the text embedding obtained from the CLIP text encoder as the prefix embedding. During **zero-shot inference**, the difficulty lies in how to obtain a prefix embedding that can match the input image and be well decoded by the decoder. The modality gap phenomenon (Liang et al., 2022b) is observed in multi-modal contrastive models which prevents us from directly taking the visual embedding as the prefix embedding. Ramesh et al. (2022) use paired data to learn a model to map the text embedding to a corresponding image embedding. Instead of learning a model, we propose a training-free mechanism to project the image embedding into the CLIP text embedding space. Combining the text decoder with the projection mechanism, we generate high-quality descriptions for given images.

Our main contributions are summarized as follows:

(1) We propose a new framework for zero-shot captioning. Our DeCap framework contains a pre-trained contrastive model (i.e., CLIP) and a lightweight visual-aware language decoder taking the CLIP embedding as input. Though our decoder is trained only on the text corpus, it can associate both the visual embedding and the text embedding, thanks to the encoded multi-modal correlation in the CLIP embedding space.

(2) We propose a training-free projection mechanism to reduce the *modality gap* in CLIP multi-modal embedding space. We incorporate a simple support memory containing embeddings of the text corpus in the pre-training stage. We project a visual embedding into the CLIP text embedding space via the support memory. Experiments show that our proposed mechanism effectively reduces the modality gap and significantly improves performance.

(3) Extensive experiments demonstrate DeCap can flexibly apply to various captioning scenarios. DeCap outperforms other zero-shot captioning methods by a large margin on image captioning benchmarks MSCOCO and NoCaps. DeCap trained on text-only data outperforms other unpaired captioning methods on MSCOCO and Flickr30k. We apply DeCap to video captioning and achieve state-of-the-art zero-shot results on MSR-VTT and ActivityNet-Captions.

## 2 RELATED WORK

**CLIP in Captioning.** Vision-language models (Radford et al., 2021; Jia et al., 2021; Yang et al., 2022) trained with a contrastive loss show impressive ability in many discriminative tasks. However, due to the absence of a text decoder during pre-training, these models can not be directly applied to generative tasks, e.g., captioning. Prior work (Mokady et al., 2021; Barraco et al., 2022; Shen et al., 2022) has applied CLIP to the image captioning task as a visual encoder. However, they ignore

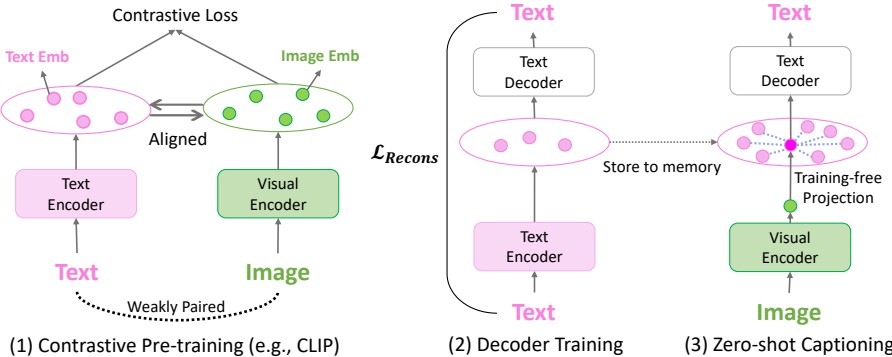

Figure 1: An overview of our framework. Our method is based on a pre-trained contrastive model CLIP containing a text encoder and a visual encoder. We first learn a text decoder to generate sentences conditioned on the CLIP text embedding. At inference, a training-free mechanism is used to project the image embedding into the text embedding space with the help of a support memory. The projected embedding is further decoded by the text decoder.

the CLIP text encoder and overlook the aligned multi-modal latent space provided by CLIP. In this work, we train a text decoder with text-only data to invert the CLIP text encoder. By leveraging CLIP multi-modal latent space, we apply CLIP to captioning tasks without additional pairwise training.

**Zero-shot Captioning.** Zero-shot captioning aims to generate image/video captions without human-annotated data. Changpinyo et al. (2021); Wang et al. (2021b); Alayrac et al. (2022) train vision-language models on noisy paired image-text data collected from the Web and evaluate on down-stream benchmarks without fine-tuning. Another line of work achieves zero-shot captioning by combining existing web-scale pre-trained models. ZeroCap (Tewel et al., 2022b) combines a multi-modal model (e.g., CLIP) with a PLM (e.g., GPT-2). In each generation step, they use CLIP to guide GPT-2 toward a desired visual direction via the proposed CLIP loss. Socratic Models (Zeng et al., 2022) use a pre-trained VLM (Gu et al., 2021) to generate prompt templates for GPT-3 (Brown et al., 2020) and then use CILP to retrieve the closest description to the image from the generated candidates. In this work, we employ CLIP for zero-shot captioning. Different from the above work using PLMs, we use text-only data to train a decoder from scratch.

**Text Reconstruction.** Prior work (Feng et al., 2019; Laina et al., 2019; Liu et al., 2021a;b) employ a text reconstruction task to train a decoder for unpaired/unsupervised captioning tasks. Lacking a well-aligned multi-modal latent space, most of these methods require complex pseudo-training or adversarial training to align the decoder and visual input. Liu et al. (2021b) construct a knowledge graph to correlate the representations of the visual and textual domains. However, this method needs a well-defined knowledge graph and a multi-label classification task to train the knowledge graph, which is difficult to apply to captioning tasks other than medical report generation. Benefiting from CLIP, on the one hand, our decoder can be directly associated with visual input by utilizing the aligned cross-modal embedding space of CLIP. On the other hand, our decoder can be trained on various text data and applied to various captioning tasks.

## 3 METHOD

Our framework is shown in Figure 1. We learn a text decoder to convert the CLIP text encoder (Sec. 3.1). This text decoder allows us to generate sentences based on the CLIP text embedding. At inference, we propose a training-free mechanism to project the image embedding into the text embedding space to reduce the modality gap between the text embedding space and image embedding space (Sec. 3.2.1). We introduce more inference strategies for comparison (Sec. 3.2.2).

### 3.1 TEXT-ONLY DECODER PRE-TRAINING

Previous approaches (Tewel et al., 2022b; Su et al., 2022; Zeng et al., 2022) employ PLMs to generate diverse sentences for zero-shot captioning. However, PLMs trained on various documents

from the webpages do not match well with captioning tasks that aim to describe visual concepts and relationships in the given image.

Instead of employing a PLM, we train a text decoder from scratch to invert the CLIP text encoder. Following recent work (Mokady et al., 2021; Wang et al., 2021b), we train our decoder using the prefix language modeling. Specifically, given a sentence $t = \{word_1, word_2, ..., word_{|t|}\}$, the prefix language model $P_\theta$ learns to reconstruct $t$ conditioned on the text embedding extracted by a fixed CLIP text encoder. We regard the text embedding as a prefix to the caption. Our objective can be described as:

$$\mathcal{L}_{Recons}(\theta) = -\frac{1}{|t|} \sum_{i=1}^{|t|} \log P_\theta(word_i | word_{<i}, E_{text}(t)), \tag{1}$$

where $E_{text}(\cdot)$ means mapping a sentence to a $\ell_2$-normalized embedding space via the CLIP text encoder. This decoder trained with text-only data in a self-supervised manner brings two benefits. On the one hand, we can control the style of the generated sentences by adjusting the source of text-only data. To generate task-specific descriptive captions, we train our decoder on text data from human-annotated image descriptions and web-collected image captions. On the other hand, this text decoder takes CLIP text embedding as the prefix embedding. The CLIP text embedding is optimized to be correlated with the CLIP image embedding, making it possible to associate the text decoder with visual input without any pairwise training.

## 3.2 INFERENCE STRATEGIES

In Sec. 3.1, we obtain a decoder that can generate descriptions conditioned on the CLIP text embedding. At inference, the question is how to use the decoder to generate descriptions given the CLIP image embedding. Due to the modality gap between CLIP image embedding space and text embedding space, it is impractical to directly take the CLIP image embedding as the prefix embedding. Ramesh et al. (2022) learn a prior model to map the text embedding to a corresponding image embedding. However, this process requires paired data for training. We propose a training-free mechanism to project the image embedding into text embedding space.

### 3.2.1 PROJECTION-BASED DECODING (PD)

Assuming that the language model $P_\theta$ is trained on a given text set $T = \{t_1, t_2, ..., t_N\}$, where $N$ denotes the size of $T$. To represent the CLIP text embedding space, we maintain a support memory $M = \{\mathbf{m_1}, \mathbf{m_2}, ..., \mathbf{m_N}\}$, where $\mathbf{m_i} = E_{text}(t_i)$. At inference, we aim to generate a caption for a given image $I$. With the help of the support memory $M$, we can project the image embedding into the text embedding space. Specifically, given the image embedding $\mathbf{v} = E_{image}(I)$, we obtain its representation in text embedding space by performing a weighted combination of all the embeddings in support memory. To obtain the weights of these text embeddings, the cosine similarity between $\mathbf{v}$ and $\mathbf{m}$ is calculated, scaled by a temperature parameter $\tau$, and normalized by a softmax function. The combined project vector $\mathbf{v}_{proj}$ is calculated as:

$$\mathbf{v}_{proj} = \sum_{i=1}^{N} w_i * \mathbf{m}_i = \sum_{i=1}^{N} \frac{\exp((\mathbf{m}_i^\top \mathbf{v})/\tau)}{\sum_{k=1}^{N} \exp((\mathbf{m}_k^\top \mathbf{v})/\tau)} * \mathbf{m}_i, \tag{2}$$

where $w_i$ is the weight of $i$-th text embedding in support memory. $\mathbf{v}_{proj}$ is a combination of CLIP text embeddings that can be used as the prefix embedding. We denote $P_\theta(\mathbf{x})$ as the auto-regressive process of generating a sentence conditioned on the prefix embedding $\mathbf{x}$. The final output can be generated by $P_\theta(\frac{\mathbf{v}_{proj}}{||\mathbf{v}_{proj}||_2})$.

This projection-based method does not require additional training. It performs well across many datasets and is flexible. The projected vector $\mathbf{v}_{proj}$ can absorb the information from text embeddings in the support memory, thereby generating diverse and accurate descriptions. On the other hand, the text data used for training and stored in support memory can be different. We can select appropriate text data to construct a new support memory according to the target domain. The image embedding will then be projected into the new text embedding space, enabling DeCap to generalize quickly to new domains without retraining.

### 3.2.2 DISCUSSION

In order to investigate the impact of our decoder and projection-based mechanism, we have included the following inference strategies for comparative analysis.

**1) CLIPRe.** We first consider a simple retrieval-based approach that does not require a decoder. This approach is mentioned in Su et al. (2022). Given the image $I$ and text set $T = \{t_1, t_2, ..., t_n\}$, CLIPRe retrieves the most relevant texts from $T$ based on the image-text similarity measured by CLIP. This process can be formulated as: $\arg\max_{t \in T} sim(E_{image}(I), E_{text}(t))$, where $sim$ denotes the cosine similarity. In all experiments, we use CLIPRe as our baseline, since it can well reflect the zero-shot performance of the original CLIP without the decoder.

**2) Visual Decoding (VD).** Considering that text embeddings and image embeddings are correlated, a simple approach is to directly use image embedding as the prefix embedding. We refer to this method as Visual Decoding. This process can be formulated as $P_\theta(E_{image}(I))$. However, across the experiments, this method does not achieve satisfying results in most scenarios, indicating that there is a modality gap between CLIP image embeddings and text embeddings.

**3) Nearest-neighbor Decoding (NND).** Another simple method is to use the nearest text embedding as the prefix embedding. Specifically, we first calculate the similarity between the image embedding $E_{image}(I)$ and the text embeddings in $M$. Then, the nearest text embedding is directly used as the prefix embedding. We refer to this method as Nearest-neighbor Decoding. This process can be formulated as $P_\theta(\arg\max_{\mathbf{m} \in M} sim(E_{image}(I), \mathbf{m}))$. Ideally, NND and CLIPRe should attain similar performance since the decoder learns to recover the origin text conditioned on the text embedding. Interestingly, across our experiments, NND achieves better performance than CLIPRe in most scenarios, suggesting that our decode may generate more descriptive sentences. Moreover, we find that the performance could be further improved by reconstructing a new text corpus using the decoder. More results and discussions can be found in Appendix B.

## 4 EXPERIMENTS

We conduct extensive experiments on captioning tasks including zero-shot image captioning, unpaired image captioning, and video captioning. We demonstrate that DeCap can efficiently achieve impressive results in diverse settings. In Sec. 4.1, we focus on zero-shot image captioning without any human annotation. In Sec. 4.2, we focus on unpaired image captioning where the images and the sentences are treated independently. In Sec. 4.3, we further apply DeCap to video captioning tasks. In Sec. 4.4, we conduct detailed ablation studies for DeCap.

**Implementation Details.** We employ a frozen pre-trained Vit-B/32 CLIP model. We adopt a 4-layer Transformer (Subramanian et al., 2018) with 4 attention heads as our language model. The size of the hidden state is 768. By default, we use all the text data in the training set to train the language model from scratch with a naive cross-entropy loss. All the text embeddings from the training corpus are stored in the support memory unless specified otherwise. At inference, the temperature $\tau$ in Eq. 2 is set to 1/150 in video captioning experiments, and 1/100 in image captioning experiments. We report the results over four standard captioning evaluation metrics: BLEU@N (Papineni et al., 2002), METEOR (Banerjee & Lavie, 2005), CIDEr (Vedantam et al., 2015), and SPICE (Anderson et al., 2016). Additionally, we use CLIP-S$^{Ref}$ (Hessel et al., 2021) and CLIP-S to measure the text-text similarity and text-image similarity, respectively. The beam search or constrained beam search (Anderson et al., 2017) is **not** used in all our results.

### 4.1 ZERO-SHOT IMAGE CAPTIONING

In this section, we conduct zero-shot image captioning using webly-collected corpora. Traditional image captioning methods rely on paired human-annotated data for training, which is difficult to obtain and limited in scale and diversity. We consider three webly-collected corpora for DeCap training: (1) **CC3M** (Sharma et al., 2018) contains three million image-description pairs collected from the web. We only use the text descriptions (CC3M-text) for training. We use one million descriptions randomly sampled from the 3M descriptions to construct the support memory. (2) **SS1M** is a webly-collected corpus specifically designed for MSCOCO caption. Feng et al. (2019) use the name of the eighty object classes in MSCOCO as keywords to crawl the descriptions from

| Methods | Pre-training stage | MSCOCO | | | | NoCaps val (CIDEr) | | | |
|---|---|---|---|---|---|---|---|---|---|
| | | B@4 | M | C | S | In | Near | Out | Overall |
| Changpinyo et al. (2021) | CC3M | - | - | - | - | 29.2 | 27.5 | 37.3 | 29.7 |
| Changpinyo et al. (2021) | CC12M | - | - | - | - | 20.7 | 24.1 | 41.6 | 27.1 |
| ZeroCap | CLIP+GPT-2 | 2.6 | 11.5 | 14.6 | 5.5 | - | - | - | - |
| CLIPRe | CLIP+CC3M-text | 4.6 | 13.3 | 25.6 | 9.2 | 23.3 | 26.8 | 36.5 | 28.2 |
| DeCap-VD | CLIP+CC3M-text | 1.2 | 10.4 | 8.1 | 5.8 | 8.4 | 8.0 | 10.2 | 8.5 |
| DeCap-NND | CLIP+CC3M-text | 5.3 | 13.7 | 27.1 | 9.1 | 24.2 | 27.1 | 37.6 | 28.8 |
| DeCap | CLIP+CC3M-text | 8.8 | 16.0 | 42.1 | 10.9 | 34.8 | 37.7 | **49.9** | 39.7 |
| DeCap | CLIP+SS1M | **8.9** | **17.5** | **50.6** | **13.1** | **41.9** | **41.7** | 46.2 | **42.7** |
| DeCap | CLIP+Book Corpus | 6.6 | 12.9 | 31.9 | 8.7 | 26.8 | 31.8 | 44.3 | 33.6 |

Table 1: Zero-shot captioning results on MSCOCO Karpathy-test split and NoCaps validation set. (In: in-domain; Near: near-domain; Out: out-of-domain; B@4: BLEU@4; M: METEOR; C: CIDEr; S: SPICE).

Shutterstock[1], resulting in 2,322,628 distinct image descriptions in total. We reuse this corpus and further remove sentences with more than fifteen words, obtaining 978,662 sentences. (3) **Book Corpus** (Zhu et al., 2015) is a large collection of free novel books. Book Corpus is often used for unsupervised pre-training of language models (Devlin et al., 2018) and we also use it to train our language decoder, but for captioning tasks. The original Book Corpus data is large and many sentences are not visual-related, which makes our decoder training inefficient. In practice, we find that the norm of CLIP text embedding can coarsely filter out some sentences that are not related to visual concepts. A sentence with a large norm is usually not visual-related. To improve training efficiency, we only keep sentences with lengths less than 15 and norms less than 10 and obtain 6,217,799 sentences for training. We use one million sentences randomly sampled from the training data to construct the support memory. In addition, we use "Attention! There is/are" as a prompt for the model trained on Book Corpus. We find that DeCap trained on Book Corpus benefits from prompt engineering, whereas DeCap trained on CC3M does not. More other prompts, results, and analyses are in Appendix F.

The following zero-shot captioning methods are compared in this study. Changpinyo et al. (2021) train a captioning model on webly-collected paired data and directly transfer it to downstream datasets without fine-tuning. **ZeroCap** (Tewel et al., 2022b) is a training-free zero-shot captioning method leveraging CLIP and GPT-2. DeCap also utilizes CLIP but trains a decoder from scratch on a webly-collected corpus. Our **DeCap** uses projection-based decoding (PD) by default. We compare it with another two inference strategies introduced in Sec. 3.2. We denote visual decoding as **DeCap-VD** and nearest-neighbor decoding as **DeCap-NND**. All these methods target zero-shot image captioning and do not use human-annotated data.

**Results.** Table 1 shows the zero-shot results on MSCOCO and NoCaps. DeCap attains a new state-of-the-art on all metrics. On NoCaps, models pre-trained on webly-collected data achieve better out-of-domain results. This is because the webly-collected data contain diverse visual concepts. On MSCOCO, DeCap pre-trained on CC3M-text outperforms ZeroCap by 27.5% in CIDEr. DeCap pre-trained on SS1M outperforms ZeroCap by 36% in CIDEr. DeCap trained on SS1M achieves better performance than trained on CC3M (CIDEr: 50.6% vs. 42.1%), indicating that the task-specific webly-collected corpus can further improve the performance of downstream datasets. Besides, DeCap trained on Book Corpus still achieves better performance than ZeroCap. Notably, both DeCap-BookCorpus and ZeroCap have not seen caption-related data.

## 4.2 UNPAIRED IMAGE CAPTIONING

To explore the potential of DeCap in more captioning scenarios, we consider the unpaired image captioning setting, where the human-annotated image-sentence pairs are treated as unpaired images and sentences. In Sec. 4.2.1, we investigate in-domain captioning where training data and test data come from the same dataset, but the training data are unpaired. In Sec. 4.2.2, we consider the cross-domain situation where training data and test data come from different distributions.

---

[1] https://www.shutterstock.com

| Method | Data | | | MSCOCO | | | | Flickr30K | | | |
|---|---|---|---|---|---|---|---|---|---|---|---|
| | P. | I. | T. | B@4 | M | C | S | B@4 | M | C | S |
| *Supervised Methods* | | | | | | | | | | | |
| BUTD | ✓ | | | 36.2 | 27.0 | 113.5 | 20.3 | 27.3 | 21.7 | 56.6 | 16.0 |
| CLIPCap | ✓ | | | 33.5 | 27.5 | 113.1 | 21.1 | - | - | - | - |
| Barraco et al. (2022) | ✓ | | | 36.0 | 27.8 | 114.9 | 20.8 | - | - | - | - |
| CLIP-VL | ✓ | | | 37.5 | 28.1 | 123.1 | 21.9 | - | - | - | - |
| *Train on unpaired data. Zero-shot inference on image-text pairs* | | | | | | | | | | | |
| UVC-VI | † | | | 22.0 | 21.4 | 72.3 | - | - | - | - | - |
| Feng et al. (2019) | | ✓ | ✓ | 18.6 | 17.9 | 54.9 | 11.1 | - | - | - | - |
| Laina et al. (2019) | | ✓ | ✓ | 19.3 | 20.2 | 61.8 | 12.9 | - | - | - | - |
| ESPER-Style | | ✓ | ✓ | 21.9 | 21.9 | 78.2 | - | - | - | - | - |
| ESPER-Free | | ✓ | | 6.3 | 13.3 | 29.1 | - | - | - | - | - |
| ZeroCap* | | | ✓ | 7.0 | 15.4 | 34.5 | 9.2 | 5.4 | 11.8 | 16.8 | 6.2 |
| Magic | | | ✓ | 12.9 | 17.4 | 49.3 | 11.3 | 6.4 | 13.1 | 20.4 | 7.1 |
| CLIPRe | | | ✓ | 12.4 | 20.4 | 53.4 | 14.8 | 9.8 | 18.2 | 31.7 | 12.0 |
| DeCap-VD | | | ✓ | 5.0 | 15.5 | 25.7 | 9.8 | 5.8 | 15.0 | 13.0 | 8.2 |
| DeCap-NND | | | ✓ | 15.3 | 21.2 | 62.9 | 15.8 | 12.9 | 17.2 | 35.2 | 10.9 |
| DeCap | | | ✓ | **24.7** | **25.0** | **91.2** | **18.7** | **21.2** | **21.8** | **56.7** | **15.2** |

Table 2: In-domain captioning results on MSCOCO and Flickr30K. "*" denotes results from Su et al. (2022). "P.", "I." and "T." denote paired data, unpaired image data and unpaired text data, respectively. †: UVC-VI is a special approach that requires image-Chinese paired data for training, and we regard it as an unpaired method here because it does not use image-English pairs.

### 4.2.1 IN-DOMAIN CAPTIONING

We compare DeCap with supervised methods and other unpaired image captioning methods. (1) Supervised methods: **BUTD** (Anderson et al., 2018) is a classic method that uses Faster R-CNN (Ren et al., 2015) to extract visual features. **CLIPCap** (Mokady et al., 2021), **CLIP-VL** (Shen et al., 2021) and Barraco et al. (2022) are recent approaches employing CLIP as the visual encoder. (2) Unpaired methods: Laina et al. (2019) and Feng et al. (2019) treat the images and captions from the MSCOCO training set as unpaired data. UVC-VI (Liu et al., 2021a) uses image-Chinese pairs (Wu et al., 2019) for training. (3) (CLIP+GPT2)-based methods: **ZeroCap** (Tewel et al., 2022b), **Magic** (Su et al., 2022) and **ESPER-Style** (Yu et al., 2022b) finetune the GPT-2 on captions from the training set. (4) **ESPER-Free** (Yu et al., 2022b) uses reinforcement learning to align multimodal inputs to language model generations. (5) **CLIPRe** is a retrieval-based baseline. (6) Our **DeCap**, **DeCap-VD** and **DeCap-NND**. Our decoder is trained on captions from the training set, and text embeddings of all the training captions are maintained in the support memory.

**Results.** Table 2 shows the results on MSCOCO and Flickr30K. Overall, DeCap outperforms recent unpaired approaches by a large margin. Especially on Flickr30K, DeCap is competitive with the supervised learning method BUTD. Two conclusions can be drawn: (1) **CLIP provides aligned multi-modal representations for captioning tasks.** Compared to the unpaired methods that use a visual concept detector to construct a multi-modal embedding space, the CLIP-based methods could achieve competitive results using only text data. (2) **Our decoder and the projection mechanism are crucial for high performance.** Compared to CLIPRe, DeCap-NND further decodes the nearest-neighbor text embeddings resulting in higher performance, indicating that our decoder can generate more descriptive sentences. DeCap-VD achieves inferior performance, demonstrating that there is a large modality gap between CLIP image embedding and text embedding, demonstrating the necessity of our projection mechanism.

### 4.2.2 CROSS-DOMAIN CAPTIONING

We evaluate the following methods on MSCOCO and Flickr30K in the cross-domain setting where the training data and testing data are from different datasets. (1) Zhao et al. (2020) generate pseudo image-text pairs for the target domain using a retrieval model trained on the source domain. (2) **Magic** (Su et al., 2022) finetunes GPT-2 on text data from the source domain. (3) **CLIPRe-S** uses text data from the source domain as galleries. (4) **DeCap** trains the decoder on text data from the

source domain. (5) **DeCap-TT** trains the decoder on text data from the source domain and uses captions from the target domain to construct the support memory.

**Results.** Table 3 shows the results. Unlike the traditional cross-domain method (Zhao et al., 2020) which relies on paired source domain data and requires training on the target domain, recent CLIP-based text-only methods require text-only data from the source domain for training. DeCap significantly outperforms other text-only methods, e.g., Magic (Su et al., 2022) and CLIPRe-S, on the cross-domain evaluation. Moreover, if the text data from the target domain is accessible, DeCap-TT significantly improves the captioning performance (e.g., CIDEr is improved from 44.4% to 63.1%) without any additional training. It simply employs text embedding from the target domain as the support memory. These results demonstrate the strong capabilities of DeCap in cross-domain generalization and the effectiveness of our projection-based decoding mechanism.

| Methods | Data | | | | MSCOCO to Flickr30K | | | | Flickr30K to MSCOCO | | | |
|---|---|---|---|---|---|---|---|---|---|---|---|---|
| | S.P. | S.T. | T.I. | T.T. | B@4 | M | C | S | B@4 | M | C | S |
| Zhao et al. (2020)* | ✓ | ✓ | ✓ | ✓ | 24.1 | 19.5 | 52.8 | - | - | - | - | - |
| Magic | | ✓ | | | 6.2 | 12.2 | 17.5 | 5.9 | 5.2 | 12.5 | 18.3 | 5.7 |
| CLIPRe-S | | ✓ | | | 9.8 | 16.7 | 30.1 | 10.3 | 6.0 | 16.0 | 26.5 | 10.2 |
| DeCap-VD | | ✓ | | | 6.5 | 13.8 | 19.1 | 7.0 | 3.6 | 13.7 | 9.4 | 6.7 |
| DeCap-NND | | ✓ | | | 12.0 | 15.5 | 28.6 | 10.1 | 7.5 | 15.9 | 28.0 | 9.6 |
| DeCap | | ✓ | | | 16.3 | 17.9 | 35.7 | 11.1 | 12.1 | 18.0 | 44.4 | 10.9 |
| DeCap-TT | | ✓ | | ✓ | 17.7 | 20.2 | 42.0 | 13.8 | 19.7 | 20.9 | 63.1 | 13.9 |

Table 3: Cross-domain image captioning evaluation. "*" means using CIDEr optimization (Rennie et al., 2017). ("S.P.": Source paired data; "S.T.": Source text data; "T.I.": Target image data; "T.T.": Target text data).

## 4.3 VIDEO CAPTIONING

In this section, we apply DeCap to the video captioning task. We conduct the experiments on MSR-VTT (Xu et al., 2016), Activity-Captions (Caba Heilbron et al., 2015), and VATEX (Wang et al., 2019). Notably, we only download 5182 raw test videos out of 6000 VATEX public test videos because some videos are unavailable. In Activity-Captions, we use ground-truth proposals following Krishna et al. (2017). We apply the same DeCap for video captioning. We consider three different data sources for decoder training: (1) **Generic corpus.** We train our decoder on Book Corpus which is a generic corpus used for unsupervised learning of language models. (2) **Image captions.** We train our decoder on captions from MSCOCO and CC3M, which are collected or annotated for image captioning tasks. (3) **Video captions**. We extract the text annotations in the training set of video captioning datasets. The former two can be viewed as the zero-shot video captioning setting without any video-related data for training.

At inference, we use a pooling mechanism on frame-level features to obtain a video-level feature. Specifically, for each proposal, we directly randomly sample $k$ frames $f_1, f_2, ..., f_k$ from the clip. We use a mean pooling mechanism on the frame-level features extracted by the CLIP image encoder to obtain a video-level feature. In all experiments, $k$ is set to 10.

**Results.** Table 4 shows the results. DeCap trained on image captions outperforms the recent zero-shot captioning approaches on standard captioning metrics and achieves competitive results on CLIP-S and CLIP-S$^{Ref}$ metrics. Notably, unlike other methods, DeCap does not directly take CLIP visual-text similarity as the optimization objective. Moreover, DeCap trained on video captions can further improve performance. These results demonstrate that DeCap can easily apply to video captioning with a simple random sampling strategy and temporal mean pooling mechanism.

## 4.4 ABLATION STUDY

**The size of training data.** A key question is how much text data we need to train the decoder from scratch. To investigate the effect of training data size, we sample different scale data from MSCOCO. At inference, we use the same support memory (full training set, 560K captions) for all experiments. The results are in Figure 2 (**left**). Overall, DeCap benefits from a large data size. Compared with training on the full set, the CIDEr score drops from 91.2% to 81.5% when using only 1% of data (5.6K captions). The result indicates that DeCap is data-efficient. It shows a promising direction in its application in data-limited scenarios.

| Methods | Setting | Metrics | | | | |
|---|---|---|---|---|---|---|
| | | B@4 | M | C | CLIP-S$^{Ref}$ | CLIP-S |
| Results on MSR-VTT test set | | | | | | |
| VNS-GRU[†] (Chen et al., 2020) | Supervised | 45.3 | 29.9 | 53.0 | 0.739 | 0.626 |
| SemSynAN[†] (Perez-Martin et al., 2021) | | 46.4 | 30.4 | 51.9 | 0.733 | 0.619 |
| UVC-VI | Trained on VATEX-Chinese (Wang et al., 2019) | 38.9 | 27.8 | 44.5 | - | - |
| ZeroCap[†] | Zero-shot | 2.3 | 12.9 | 5.8 | 0.739 | 0.710 |
| MAGIC[†] | | 5.5 | 13.3 | 7.4 | 0.628 | 0.566 |
| Tewel et al. (2022a)[†] | | 3.0 | 14.6 | 11.3 | 0.785 | **0.775** |
| DeCap-BookCorpus | | 6.0 | 12.7 | 12.3 | 0.772 | 0.719 |
| DeCap-CC3M | | 6.2 | 14.9 | 15.0 | **0.792** | 0.736 |
| DeCap-COCO | | **14.7** | **20.4** | **18.6** | 0.761 | 0.697 |
| CLIPRe-MSR | MSR-VTT text only | 10.2 | 18.8 | 19.9 | **0.835** | **0.852** |
| DeCap-VD-MSR | | 5.9 | 16.3 | 10.2 | 0.765 | 0.722 |
| DeCap-NND-MSR | | 13.1 | 20.2 | 24.4 | 0.805 | 0.771 |
| DeCap-MSR | | **23.1** | **23.6** | **34.8** | 0.823 | 0.770 |
| Results on ActivityNet- Caption $ae\text{-}test$ (Lei et al., 2020) | | | | | | |
| Reasoner (Liang et al., 2022a) | Supervised | 12.5 | 16.4 | 30.0 | - | - |
| PDVC (Wang et al., 2021a) | | 11.8 | 15.9 | 27.3 | - | - |
| DeCap-BookCorpus | Zero-shot | 0.4 | 4.4 | 10.0 | 0.734 | 0.750 |
| DeCap-CC3M | | 0.7 | 5.3 | 12.4 | **0.761** | **0.814** |
| DeCap-COCO | | **1.1** | **6.6** | **15.0** | 0.727 | 0.753 |
| CLIPRe-ACT | ActivityNet-Captions text only | 1.4 | 8.2 | 15.1 | **0.830** | **0.871** |
| DeCap-VD-ACT | | 1.1 | 6.6 | 10.2 | 0.682 | 0.712 |
| DeCap-NND-ACT | | 1.9 | 8.3 | 15.5 | 0.745 | 0.775 |
| DeCap-ACT | | **2.3** | **9.4** | **20.6** | 0.767 | 0.797 |
| Results on VATEX public test set | | | | | | |
| VaTeX (Wang et al., 2019) | Supervised | 28.4 | 21.7 | 45.1 | - | - |
| DeCap-BookCorpus | Zero-shot | 4.1 | 9.9 | 11.8 | 0.761 | 0.731 |
| DeCap-CC3M | | 7.3 | 12.6 | 18.4 | **0.804** | **0.802** |
| DeCap-COCO | | **13.1** | **15.3** | **18.7** | 0.769 | 0.755 |
| CLIPRe-VATEX | VATEX-Captions text only | 11.1 | 17.0 | 27.1 | **0.835** | **0.877** |
| DeCap-VD-VATEX | | 7.4 | 12.9 | 13.8 | 0.732 | 0.733 |
| DeCap-NND-VATEX | | 14.8 | 18.1 | 32.4 | 0.809 | 0.811 |
| DeCap-VATEX | | **21.3** | **20.7** | **43.1** | 0.834 | 0.824 |

Table 4: Video captioning evaluation results. "†" denotes the result from Tewel et al. (2022a). DeCap-BookCorpus, DeCap-CC3M, DeCap-COCO, DeCap-MSR, DeCap-ACT and DeCap-VATEX denote the model is trained on text data from Book Corpus, CC3M, MSCOCO, MSR-VTT, Activity-Captions, and DeCap-VATEX, respectively.

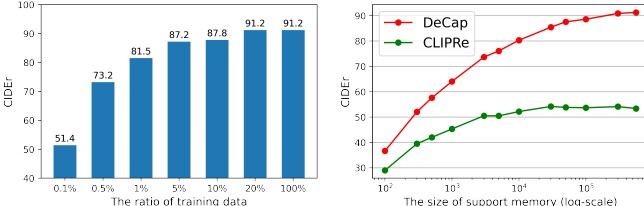

Figure 2: Ablation study on the training data size (**left**) and the support memory size (**right**).

**The size of the support memory.** To investigate the effect of support memory size, we first train the language model on the full training set (560K captions). At inference, we randomly sample different ratio text embeddings as the support memory. The results are in Figure 2 (**right**). Overall, DeCap and CLIPRe both benefit from a large support memory. Moreover, when using only 1% data as the support memory, the performance drops slightly (3.8% performance drop in CIDEr). It indicates that we can maintain a relatively small support memory to achieve competitive results with acceptable storage and computation costs. Additionally, we provide a filtering strategy to reduce the number of support embeddings in Appendix E. We visualize the support memory and the projection embedding in Appendix G. We add an inference speed analysis to Appendix D.

## 5 CONCLUSION

We propose a simple framework for zero-shot captioning and introduce a lightweight visual-aware language decoder that is both data-efficient and computation-efficient. We propose a training-free mechanism to project the visual embedding to text embedding space, significantly reducing the modality gap issue. By combining the decoder with the projection mechanism, we significantly outperform existing zero-shot methods, establishing a new state-of-the-art in MSCOCO, MSR-VTT, and ActivityNet-Captions. In the future, our DeCap framework may be adapted to other zero-shot text generation problems, e.g., visual dialog.

## ACKNOWLEDGMENTS

This work is supported by National Key R&D Program of China under Grant No. 2020AAA0108800. This work is partially supported by the Fundamental Research Funds for the Central Universities (No. 226-2022-00051).

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

## A  MORE IMPLEMENTATION DETAILS

We employ a frozen pre-trained Vit-B/32 CLIP model as our cross-modal feature extractor. We adopt a lighting 4-layer Transformer (Subramanian et al., 2018) with 4 attention heads as our decoder (hidden state size 768) following the details (Radford et al., 2019). A linear layer trained with the decoder is used to project the CLIP embedding from 512 to 768 dimensions. The training data size and hyper-parameter for different datasets are summarized in Table 5.

|  | MSCOCO | Flickr30K | CC3M | SS1M | MSR-VTT | Activity-Captions | Book Corpus |
|---|---|---|---|---|---|---|---|
| Training size | 560K | 30K | 3M | 978K | 140K | 37K | 6M |
| Training steps | 40K | 20K | 200K | 150K | 20K | 8K | 400K |
| Warmup steps | 2K | 2K | 2K | 2K | 2K | 1K | 2K |
| Batch size |  |  |  | 128 |  |  |  |
| Learning rate |  |  |  | $1e^{-5}$ |  |  |  |
| Label smoothing |  |  |  | 0.1 |  |  |  |
| Optimizer |  |  | AdamW (Loshchilov & Hutter, 2018) with default hyperparameters |  |  |  |  |

Table 5: Training data size and hyper-parameter

## B  DISCUSSION ABOUT THE RECONSTRUCTION

In Sec. 3.2.2, we introduce the CLIPRe and Nearest-neighbor Decoding (NND) method. Given an image and its CLIP image embedding, both CLIPRe and NND first retrieve the most relative text embedding $\mathbf{m_t}$ in the support memory according to the image-text cosine similarity. CLIPRe then adopts the original sentence $t$ of the $\mathbf{m_t}$ as the caption. NND uses the decoder to generate a sentence $t^*$ conditioned on $\mathbf{m_t}$. Ideally, the generated sentence $t^*$ should be the same as the original sentence $t$, because the decoder learns to reconstruct $t$ conditioned on the $\mathbf{m_t}$. However, according to the experiments in Sec. 4.2.1, we find NND outperforms CLIPRe in most metrics. To figure out the reason, we conduct the following experiments.

We first train our decoder on the MSCOCO training set with Eq. 1. To investigate the effect of the text decoder, we construct a new corpus $T^* = \{t_1^*, t_2^*, ..., t_N^*\}$, where $t^* = P_\theta(E_{text}(t))$, $t$ is the original sentence in MSCOCO training set and $t^*$ is the reconstructed sentence. We adopt this new corpus as the support memory.

Table 6 shows the results. The reconstructed dataset improves the performance of CLIPRe on all metrics, especially on CIDEr, from 53.4% to 63.6% (+10.2%). DeCap adopting the new corpus as the support memory could further improve the CIDEr score to 95.1% (+3.9%). The result demonstrates that the sentences generated by our method can better describe the images in MSCOCO. We think the reason is that our decoding process has a denoising effect, which can remove some outliers captions in the training set. Another open question here is whether such a denoised dataset can improve the performance of other fully supervised methods. We leave this as our future work.

|  | B@4 | M | C | S |
|---|---|---|---|---|
| CLIPRe | 12.4 | 20.4 | 53.4 | 14.8 |
| CLIPRe-recons | 14.9 (+2.5) | 21.5 (+1.1) | 63.6 (+10.2) | 16.2 (+1.4) |
| DeCap | 24.7 | 25.0 | 91.2 | 18.7 |
| DeCap-recons | 26.5 (+1.8) | 24.9 (-0.1) | 95.1 (+3.9) | 18.6 (-0.1) |

Table 6: Results on MSCOCO Karpathy-test split. CLIPRe-recons and DeCap-recons denote using the reconstructed corpus as the support memory.

## C  PRETRAINING-FINETUNING.

An interesting question is whether DeCap can benefit from the pretraining-finetuning paradigm. Table 7 shows the results. Notably, we only use text data for training in both pre-training and fine-tuning. Compared to training on MSCOCO, the model trained on CC3M achieves better performance in the out-of-domain case, improving the CIDEr metric from 25.8% to 48.7%. This is

| Pre-training data | Fine-tuning data | Memory data | CIDEr | | | |
|---|---|---|---|---|---|---|
| | | | in | near | out | overall |
| MSCOCO | - | MSCOCO | 65.2 | 47.8 | 25.8 | 45.9 |
| CC3M | - | CC3M | 34.7 | 35.9 | **48.7** | 38.3 |
| CC3M | MSCOCO | MSCOCO | **72.7** | **61.9** | 43.9 | 58.2 |
| CC3M | - | MSCOCO | 70.1 | 60.4 | 44.5 | **58.6** |

Table 7: Results of DeCap on NoCaps validation split. We only use the text data for both pre-training and fine-tuning.

because CC3M covers more diverse classes than MSCOCO. By fine-tuning the pre-trained model on MSCOCO, we find that the overall performance is greatly improved, obtaining an overall CIDEr of 58.2%. It indicates that our method benefits from the pretraining-finetuning paradigm. By directly changing the support memory without fine-tuning, DeCap achieves comparable performance as fine-tuning. It suggests that our method can be easily adapted to new domains without training, requiring only some text data from new domains.

## D    THE INFERENCE SPEED

Table 8 shows the inference speed of DeCap. Decap is 113x faster than ZeroCap. Because DeCap does not involve gradient updates and multiple text encoder forwards during the inference. Besides, the decoder used in DeCap is more lightweight compared to the GPT-2 employed in ZeroCap. It is worth mentioning that the time cost of embedding projection is negligible compared to image encoding and text decoding.

| | Image encoding (CLIP image encoder) | Embedding projection (1M support memory) | Language decoding | All | FPS |
|---|---|---|---|---|---|
| ZeroCap | 32.68 ms | - | 11285.36 ms | 11318.04 ms | 0.088 |
| DeCap | 31.75 ms | 0.38 ms | 68.54 ms | 100.67 ms | 9.933 |

Table 8: The inference speed of ZeroCap and DeCap. The experiment is conducted on a single Nvidia RTX2080Ti GPU. Both DeCap and ZeroCap do not use the beam search. We report the average time cost of captioning 100 images with batch size 1.

## E    AN EFFICIENT STRATEGY TO REDUCE THE NUMBER OF SUPPORT EMBEDDINGS

To make DeCap more practical, we provide a method that does not degrade DeCap performance but can significantly reduce the number of support embeddings. In the original DeCap, we randomly sample sentences from the training set to construct the support memory. However, the semantics between sentences is highly repetitive. A simple but effective method is to filter the features in the support memory according to the cosine similarity. Specifically, given a text feature and the existing support memory, if the maximum cosine similarity between the feature and the support memory is greater than a threshold, the feature will not be stored in the support memory. We set the threshold to 0.8 and construct a new support memory with the filtering strategy. Table 9 shows that this strategy can significantly reduce the number of support embeddings from 1M to 0.14M and thus reduce the additional memory cost from 1.02GB to 0.14GB without performance degradation.

| Similarity filter | The number of support embeddings | Additional memory cost | CIDEr |
|---|---|---|---|
| False | 1M | 1.02GB | 42.2 |
| False | 0.14M (randomly sampled from 1M) | 0.14GB | 38.2 (-4.0) |
| True | 0.14M (Filtering from 1M) | 0.14GB | 42.3 (+0.1) |

Table 9: The result of filtering strategy. We use the same 1M sentences in this experiment.

## F  PROMPT ENGINEERING

Prior work (Tewel et al., 2022b; Wang et al., 2021b) found that a prefix prompt "a picture of" improves the quality of decoded captions. We study the effect of the prompt on our special decoder trained with a text reconstruction task. We consider two decoders trained on CC3M-text and Book Corpus, respectively. At inference, we take the "prefix embedding + prompts" as the input of the decoder. We test a set of prompts as shown in Table 10. The results show that the decoder trained on Book Corpus benefits from the prompt engineering, while the decoder trained on CC3M hurts from the prompt engineering in most cases. Although CC3M is a dataset collected automatically from the Web, it is well-filtered by some human-designed strategies. Therefore, most of the text data in CC3M are caption-related, and the redundant prompt design will destroy its original text structure, resulting in performance degradation. BookCorpus is a popular large-scale text corpus, especially for unsupervised learning of language models. Most of the sentences in Bookcorpus are not originally intended to describe pictures. A well-designed prompt can allow the decoder to generate sentences that match the captioning task.

| Prompt | DeCap-BookCorpus | DeCap-CC3M |
|---|---|---|
| None | 21.8 | 42.1 (+0.0) |
| "A photo of" | 20.4 (-1.4) | 38.3 (-3.8) |
| "A picture of" | 20.8 (-1.0) | 36.8 (-5.3) |
| "There is/are" | 27.1 (+5.3) | 40.7 (-1.4) |
| "See! There is/are" | 30.4 (+8.6) | 40.9 (-1.2) |
| "Attention! There is/are" | 31.9 (+10.1) | 42.2 (+0.1) |

Table 10: Zero-shot captioning results on MSCOCO Karpathy split with different prompts.

## G  VISUALIZATION OF THE EMBEDDINGS

Figure 3 shows the support embeddings and category embeddings from MSCOCO. The support embeddings from the clip text encoder are divided into different clusters according to the semantics. In Figure 4(a), we sample 500 image-text **pairs** from the MSCOCO training set and visualize their embeddings extracted by the CLIP encoder. As we can see, there is a clear modality gap between CLIP text embeddings and image embeddings. Figure 4(b) and Figure 4(c) show that the projection method can effectively reduce this modality gap. Figure 4(d) shows that the projected embedding is close to the embeddings of human-labeled captions in the latent space. Besides, compared to CLIPRe embedding, the projected embedding is more central, indicating that the projected embedding absorbs the information of the support embeddings and nicely preserves the visual information.

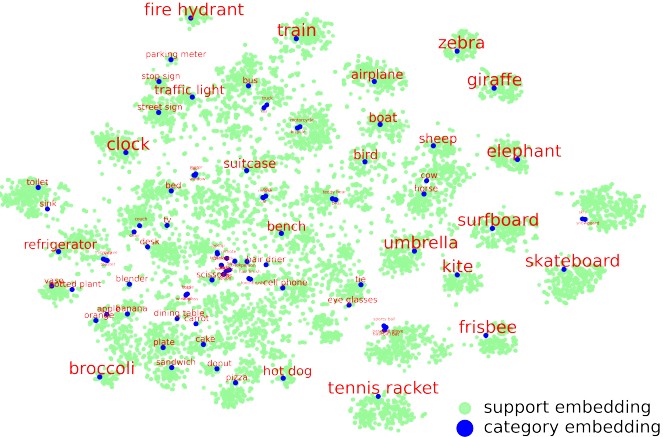

Figure 3:    Visualization of support embeddings and category embeddings from MSCOCO in 2D space by t-SNE (Van der Maaten & Hinton, 2008). We randomly sample 10,000 embeddings from the support memory for visualization.

Figure 4: Visualization of embeddings in 2D space by t-SNE. We construct the support memory using text embeddings from MSCOCO training set and randomly sample 500 embeddings from the support memory for visualization.

## H  EXAMPLES OF GENERATED CAPTIONS

We visualize the generated captions of some images from the MSCOCO Karpathy-test split in Figure 5. We show the captions generated by the DeCap model trained on MSCOCO and CC3M. The captions from DeCap-MSCOCO and DeCap-CC3M have visible style differences.

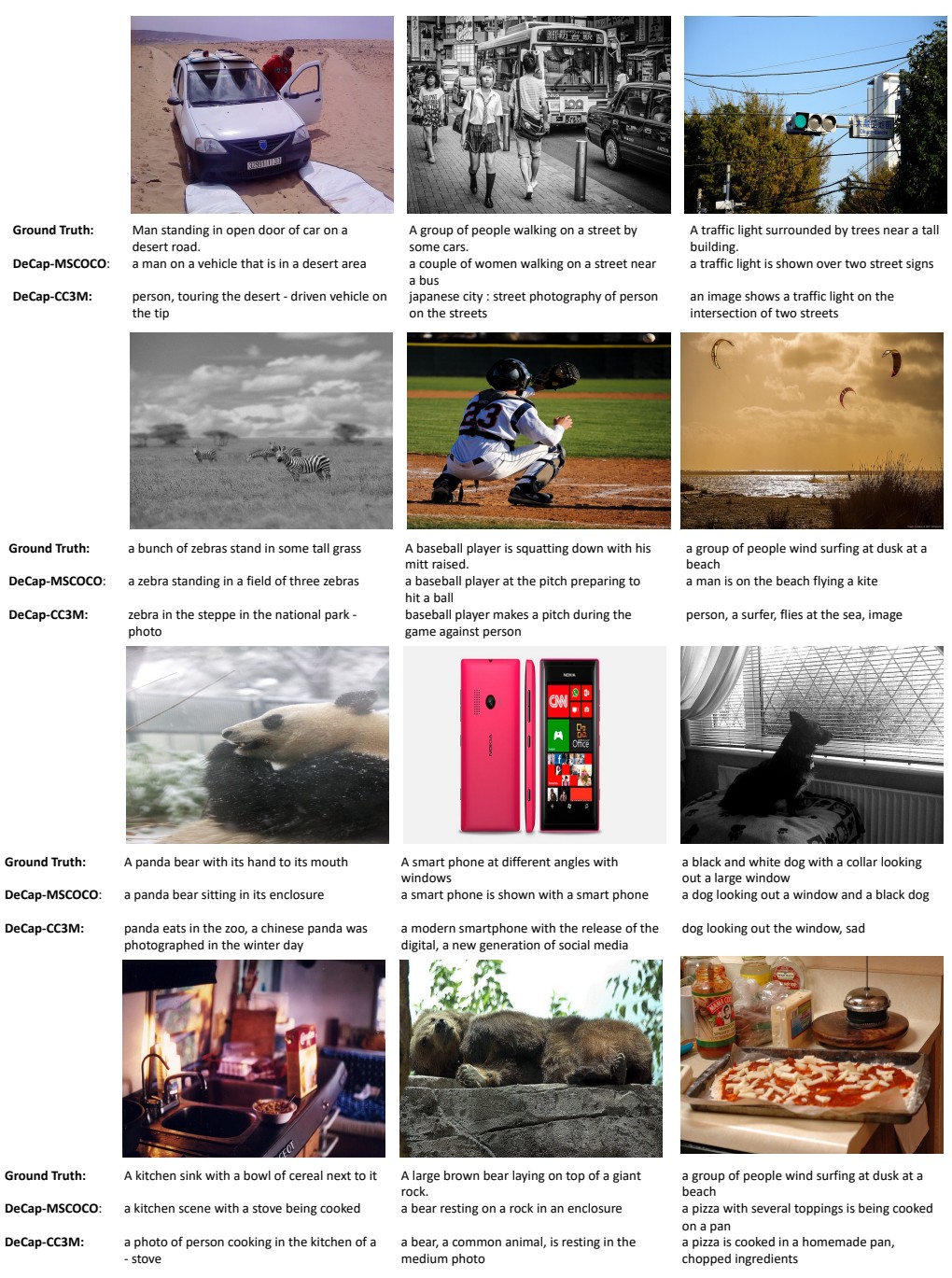

Figure 5: Generated captions for images from the MSCOCO Karpathy-test split. DeCap-MSCOCO and DeCap-CC3M denote DeCap trained on the MSCOCO training set and CC3M training set, respectively.

