# OpenReview forum: "DeCap: Decoding CLIP Latents for Zero-Shot Captioning via Text-Only Training"
_ICLR.cc/2023/Conference — ICLR 2023 poster_

### Official Review · Reviewer_CANm · 2022-10-26

**Confidence:** 4
**Correctness:** 3
**Technical Novelty And Significance:** 3
**Empirical Novelty And Significance:** 3
**Recommendation:** 6

**Clarity, Quality, Novelty And Reproducibility:**

The overall clarity and quality are good. The novelty is limited, an important reference should be discussed. The reproducibility of this paper is good: key resources (i.e., code and data) will be available upon publication and sufficient details of the experimental setup are described such that the researchers could be able to reproduce the main results. However, the fairness of the experiments should be clarified.

**Strength And Weaknesses:**

Strengths:
1. The explored problem is valuable. The proposed approach is sound. The proposed training-free projection mechanism is interesting.
2. The proposed approach can significantly outperform previous zero-shot captioning methods.

Weaknesses:
1. The idea of the proposed approach is similar to the existing work [1]. Meanwhile, the performances achieved by [1] are better than in this work. However, this work neither cites nor compares with [1]. Although it's acceptable that the proposed approach underperforms the previous work [1] in some scenarios, I still recommend the authors discuss the differences between this work and previous work [1] in Introduction and Related Work. It can help the readers better understand the contribution of this work.

2. The fairness of the experiments should be clarified.
- Compared with previous works, the approach proposed in this work adopts an external strong pre-training model, i.e., CLIP, and several external large-scale datasets, e.g., CC3M, to perform the zero-shot tasks. So I am wondering how many contributions of the achieved best results are brought by the proposed approach, instead of the CLIP and the additional datasets.
- Could you report the performance of the proposed approach without the CLIP and/or CC3M?

3. Some experiments are missing.
- Could you perform a detailed ablation study of Equation (2)? That is, what are the performances without m_i^T v, i.e., ∑w_i m_i? What are the performances of P(v_proj) instead of P(v_proj/||P(v_proj)||_2)
- What are the performances of the fully-supervised DECAP model?
- Could you visualize the learned representations, e.g., the visual representations, the projected visual representations, and the textual representations? It can better intuitively prove your arguments that the proposed approach can alleviate the modality gap problem.

Reference:
[1] Aligning Source Visual and Target Language Domains for Unpaired Video Captioning. TPAMI, 2021

**Summary Of The Paper:**

This work aims to utilize the encoded vision-and-language correlation in the trained CLIP embedding space to propose a DeCap method to perform zero-shot captioning. In implementations, this work first constructs an auto-encoder language model to learn to generate the given sentence based on the text embedding encoded by the off-the-shelf pre-trained CLIP. Then, this work further proposes a training-free projection mechanism to project the CLIP's visual embedding into the CLIP's text embedding space, which will be used in the trained auto-encoder language model to generate the final captions. The experiments on the image captioning and video captioning benchmark datasets show that the DeCap can significantly outperform previous methods under comparable settings.

**Summary Of The Review:**

Overall, the paper has provided extensive experiments to show that the proposed approach is effective in performing zero-shot captioning. However, the fairness of the experiments should be clarified.

----After Rebuttal---

I've read the rebuttal and the reviews of fellow reviewers. Many thanks for the detailed response. I have no further questions regarding this paper.

---

> ### Author Response · Authors · 2022-11-19
> **Reply to Reviewer 6 CANm (Part 1)**
>
>
> > Q1: The idea of the proposed approach is similar to the existing work [1]. Meanwhile, the performances achieved by [1] are better than in this work. However, this work neither cites nor compares with [1]. Although it's acceptable that the proposed approach underperforms the previous work [1] in some scenarios, I still recommend the authors discuss the differences between this work and previous work [1] in Introduction and Related Work. It can help the readers better understand the contribution of this work.
>
> Ans:  Thank you for your comments. We have added the comparison with [1] in our revision.
> [1] mainly studies how to **translate a captioning model** from the source language to the target language while DeCap studies how to apply a contrastive model (i.e., CLIP) to generative tasks (i.e., captioning). The differences between [1] and our work are below:
>
> (1) Data: [1] requires **human-labeled** **paired data** for training (e.g., video-Chinese pairs from VATEX). DeCap just needs **text-only** data which can be **collected from the Web** (e.g., CC3M, SS1M and BookCorpus) or from human annotations (e.g., MSCOCO).
>
> (2) Details: [1] proposes a *Visual Injection Module (VIM)* to project the visual embedding to the target language domain. Though it may look similar to our projection method, the difference is that VIM **requires pseudo-paired data for training** while our **projection method is training-free**.
>
> (3) Performance: Since the data used by the two methods are quite different as mentioned before, it may be unfair to directly compare the evaluation metrics between DeCap and [1]. Considering that [1] is also a good solution for captioning without target-paired data, we have added the results of [1] to Table 2 and Table 4 as an unpaired method for reference.

---

> > ### Author Response · Authors · 2022-11-19
> > **Reply to Reviewer 6 CANm (Part 2)**
> >
> >
> > > Q2: The fairness of the experiments should be clarified.
> >
> > > a) Compared with previous works, the approach proposed in this work adopts an external strong pre-training model, i.e., CLIP, and several external large-scale datasets, e.g., CC3M, to perform the zero-shot tasks. So I am wondering how many contributions of the achieved best results are brought by the proposed approach, instead of the CLIP and the additional datasets.
> >
> > Ans:
> >
> > 1. External datasets: For a fair comparison, we train DeCap on the same datasets as compared methods across the experiments.    (1) In Table 1, both of DeCap and [3] are trained on CC3M. Notably, Zero-Cap is a training-free method and adopts GPT-2 as a text generator which is trained on a large generic corpus. For a fairer comparison with ZeroCap, we add a new experiment to train DeCap on BookCorpus, which is also a general-purpose corpus. The results show that DeCap-BookCorpus still outperforms ZeroCap. (2) In Table 2 and Table 3, DeCap and other methods use images/captions from MSCOCO and Flickr30K for training. (3) In Table 4, we report the results of DeCap trained on different datasets (e.g., DeCap-COCO, DeCap-CC3M, DeCap-MSR, and DeCap-ACT).
> >
> > 2. External model: (i) **We provided some CLIP-based methods for fair comparison in the experiments**. Some of them even adopt a pre-trained GPT-2. (ii) **We provided three baselines based on CLIP as discussed in Section 3.2.2**.
> >
> > 3. The contributions of the proposed approach: We provided three baselines (e.g., CLIPRe, DeCap-VD and DeCap-NND) in Section 3.2.2 to investigate the effectiveness of the proposed decoder and projection method.
> >
> >    The CLIPRe is a retrieval-based method **without projection and decoding**. DeCap-VD directly takes the CLIP image embedding as the input of the decoder **without support memory.** DeCap-NND retrieves the most similar text embedding as the input of the decoder without projection.  DeCap-VD achieves inferior results in all cases because of the **modality gap** between CLIP image embeddings and text embeddings. We have visualized the modality gap in Appendix F. DeCap-NND achieves better performance than CLIPRe in most cases, indicating the **effectiveness of the decoder** (More results and discussions about the CLIPRe and DeCap-NND are in Appendix B). DeCap outperforms three baselines and other zero-shot methods by a large margin, indicating that DeCap is a strong framework for zero-shot/unpaired captioning.
> >
> >    | Methods   | Support memory | Decoder | Projection | CIDEr |
> >    | --------- | -------------- | ------- | ---------- | ----- |
> >    | CLIPRe    | &check;        | &cross; | &cross;    | 53.4  |
> >    | DeCap-VD  | &cross;        | &check; | &cross;    | 25.7  |
> >    | DeCap-NND | &check;        | &check; | &cross;    | 62.9  |
> >    | DeCap     | &check;        | &check; | &check;    | 91.2  |
> >
> > > b) Could you report the performance of the proposed approach without the CLIP and/or CC3M?
> > >
> >
> > Ans:
> >
> > 1. Without CLIP: As our title indicates, we aim to explore the latent space from the CLIP model for the captioning task. To achieve this goal, our DeCap requires an aligned image-text latent space (e.g., CLIP latent space) and some text data (e.g., CC3M-text). We follow the recent literature [2] on zero-shot captioning. Recent advances in zero-shot captioning adopt CLIP to achieve strong performances. We follow these practices and further exploit the CLIP latents with our learnable decoder and projection-based decoding.
> > 2. Without CC3M: We reported the results of DeCap trained on different datasets, including CC3M and SS1M in Section 4.1, MSCOCO and Flickr30K in Section 4.2, and video captions from MSR-VTT and Activity-net in Section 4.3.  Besides, we have added a new experiment that trains the decoder on **BookCorpus** which is a generic corpus for unsupervised learning of language models. The results demonstrate that DeCap still works **without any caption-related corpus** for training.

---

> > > ### Author Response · Authors · 2022-11-19
> > > **Reply to Reviewer 6 CANm (Part 3)**
> > >
> > >
> > > > Q3: Some experiments are missing.
> > > >
> > >
> > > > a): Could you perform a detailed ablation study of Equation (2)? That is, what are the performances without m_i^T v, i.e., ∑w_i m_i? What are the performances of P(v_proj) instead of P(v_proj/||P(v_proj)||_2)
> > > >
> > >
> > > Ans: (1) We find that $m_i^T v$ has a minor effect on the results. We have removed this term in Eq. 2 and updated the results of DeCap in all experiments. (2) Projected embedding normalization: In the training stage, the prefix embedding of language model P is normalized, so we keep the same at inference. The CIDEr drops 3.4% without normalization (without $m_i^Tv$ here).
> > >
> > > |       | With normalization | Without normalization |
> > > | ----- | ------------------ | --------------------- |
> > > | CIDEr | 91.2               | 87.8                  |
> > >
> > > > b) What are the performances of the fully-supervised DECAP model?
> > >
> > > Ans: (1) We try to directly learn a mapping from CLIP image embeddings to CLIP text embeddings using **paired image-text data** from MS-COCO with an MSE loss. Specifically, we first extract the visual embeddings of images through the frozen CLIP image encoder. A two layers MLP with batch normalization is then learned to map the visual embeddings to its paired text embeddings with an MSE loss. We use all the image-text pairs from the MSCOCO training set to train the mapping network. However, the result is lower than the proposed projection method (84.3% vs. 91.2%, CIDEr score) which requires text-only data. (2) We do not perform DeCap in a fully supervised setup (end-to-end training), because DeCap is designed especially for zero-shot captioning. The idea of adapting CLIP models to image captioning tasks in a fully-supervised setup has been explored in prior work (e.g., [4] and [5]).
> > >
> > > > c) Could you visualize the learned representations, e.g., the visual representations, the projected visual representations, and the textual representations? It can better intuitively prove your arguments that the proposed approach can alleviate the modality gap problem.
> > > >
> > >
> > > Ans: Thanks for your suggestion. We add the visualization of CLIP embeddings, the support memory and the projection to Appendix G. From the visualization we can observe: (i) There is a clear modality gap between CLIP image embeddings and CLIP text embeddings.  (ii) The projection method can well alleviate the modality gap. (iii) The projected embedding is close to the ground truth (human-labeled captions), indicating that the projected embedding nicely preserves the visual information.
> > >
> > > Ref:
> > >
> > > [1] Liu, Fenglin, et al. "Aligning source visual and target language domains for unpaired video captioning."  (TPAMI 2021)
> > >
> > > [2] Tewel, Yoad, et al. "ZeroCap: Zero-Shot Image-to-Text Generation for Visual-Semantic Arithmetic." (CVPR 2022)
> > >
> > > [3] Changpinyo, Soravit, et al. "Conceptual 12m: Pushing web-scale image-text pre-training to recognize long-tail visual concepts."  (CVPR 2021)
> > >
> > > [4] Barraco, Manuele, et al. "The Unreasonable Effectiveness of CLIP Features for Image Captioning: An Experimental Analysis." (CVPR 2022)
> > >
> > > [5] Shen, Sheng, et al. "How Much Can CLIP Benefit Vision-and-Language Tasks?."  (ICLR 2022)

---

> > > > ### Comment · Reviewer_CANm · 2022-12-03
> > > > **Thank you**
> > > >
> > > > ----After Rebuttal---
> > > > Many thanks for the efforts and the detailed response. I have no further questions regarding this paper.

---

> > > > > ### Author Response · Authors · 2022-12-04
> > > > > **Thank you**
> > > > >
> > > > > We are happy to read that we answered your questions. Thank you for your suggestions helped us to improve our manuscript.

---

### Official Review · Reviewer_1suB · 2022-10-27

**Confidence:** 4
**Correctness:** 2
**Technical Novelty And Significance:** 2
**Empirical Novelty And Significance:** 2
**Recommendation:** 6

**Clarity, Quality, Novelty And Reproducibility:**

* This paper is well-written and easy to follow.
* This work is simple but effective.

**Strength And Weaknesses:**

Strength:
* This paper is well-written and easy to follow. The paper provides sufficient technical details for readers to understand.
* The results are very promising. The experiments on both zero-shot image captioning and zero-shot video captioning settings show that DeCap can outperform previous state-of-the-art methods.
* The text reconstruction module in decoder training is simple and effective. And the projection-based decoding is alleviating the modality gap phenomenon.

Weakness:
* Lack of important references and comparations. The idea of text reconstruction is not the first work in the captioning area. The authors should compare the differences between the DeCap and [1][2].
* The introduction claims that “the inference speed of these methods is slow because each word generation involves a CLIP text encoder forward”. It is necessary to conduct inference speed experiments to make a claim more convincing.
* It is not clear how the training-free projection mechanism can be beneficial to the modality gap. I suggest that use T-SNE to visualize the distribution of the v_proj, M, and v, it will be very interesting.


[1] Auto-Encoding Knowledge Graph for Unsupervised Medical Report Generation. NeurIPS 2021.
[2] Aligning Source Visual and Target Language Domains for Unpaired Video Captioning. TPAMI 2022.


**Summary Of The Paper:**

This paper focuses on the problem that the usage of GPT-2 for zero-shot captioning is not reasonable due to the task discrepancy between captioning and language modeling. To this end, this paper proposes the DeCap, which aims to decode sensible visual descriptions from the CLIP multi-modal embedding space. Specifically, the DeCap contains a pretrained contrastive model and a lightweight visual-aware language decoder taking the CLIP embedding as input. Expensive experiments on the MSCOCO, Flickr30k, MSR-VTT, and ActivityNet-Captions datasets show that DeCap can outperform previous state-of-the-art methods.

**Summary Of The Review:**

Overall, I think this work is simple but effective, and shows promising results for zero-shot captioning. However, this paper missed some important references and some claims are imprecise. So, I tend to not fully accept the current version of this paper.

---

> ### Author Response · Authors · 2022-11-19
> **Reply to Reviewer 5 1suB**
>
>
> > Q1: Lack of important references and comparations. The idea of text reconstruction is not the first work in the captioning area. The authors should compare the differences between the DeCap and [1] [2].
>
> Ans: Thanks for sharing these two references. We have added a comparison between DeCap and other work (e.g., [1],[2],[3], and [4]) using text reconstruction for unsupervised/unpaired captioning to the related work. It is worth mentioning that we did not claim that "DeCap is the first work to use text reconstruction for captioning".
>
> 1. Comparison with [1]: Both of DeCap and [1] take a text reconstruction task to train a text decoder for captioning task. The key differences are that: **(1) The multi-modal embedding space**: [1] constructs a knowledge graph to correlate the representations of the visual and textual domains. However, this method needs a well-defined knowledge graph and a multi-label classification task to train the knowledge graph, which is **difficult to apply to captioning tasks other than medical report generation**. Benefiting from the pre-trained CLIP, DeCap can directly associate the decoder with visual input by utilizing the aligned cross-modal embedding space of CLIP without any additional training. **(2) The data used for training:** [1] mainly focus on medical report generation task and requires independent sets of medical images and medical reports for training which may still be difficult to collect. Benefiting from CLIP which is pre-trained on large-scale web-collected data, DeCap can be trained on various text data (e.g., web-collected data, human annotations, and a generic corpus) and applied to various captioning tasks.
> 2. Comparison with [2]: [2] studies how to **translate a captioning model** from the source language to the target language while DeCap studies how to apply a contrastive model (i.e., CLIP) to generative tasks (i.e., captioning). [2] requires **human-labeled** **paired data** for training (e.g., video-Chinese pairs from VATEX). DeCap just needs **text-only** data which can be **collected from the Web** (e.g., CC3M, SS1M and BookCorpus) or from human annotations (e.g., MSCOCO). [2] proposes a *Visual Injection Module (VIM)* to project the visual embedding to the target language domain. Though it may look similar to our projection method, the difference is that VIM **requires pseudo-paired data for training** while our **projection method is training-free**.
>
> > Q2: The introduction claims that “the inference speed of these methods is slow because each word generation involves a CLIP text encoder forward”. It is necessary to conduct inference speed experiments to make a claim more convincing.
>
> Ans:  We have added the analysis of the inference time of DeCap to Appendix D. The results show that DeCap is **113x faster** than ZeroCap and the **time cost of embedding projection is negligible** compared to the image encoding and text decoding. We conduct this experiment on a single Nvidia RTX2080Ti GPU. Both DeCap and ZeroCap do not use the beam search. We report the average time cost of captioning **100 images** with batch size 1. The inference speed of DeCap is 9.933 FPS which is 113x faster than ZeroCap (0.088 FPS) which involves gradient updates and multiple CLIP text encoder forwards during inference. Besides, the time cost of the proposed projection is only 0.38 ms, which is **negligible** compared to image encoding (31.75 ms) and language decoding (68.54 ms).
>
> | Methods | Image encoding | Projection (1M) | Language decoding | All         | FPS   |
> | ------- | -------------- | -------------------- | ----------------- | ----------- | ----- |
> | ZeroCap | 32.68 ms       | -                    | 11285.36 ms       | 11318.04 ms | 0.088 |
> | DeCap   | 31.75 ms       | **0.38 ms**          | 68.54 ms          | 100.67 ms   | 9.933 |
>
> > Q3: It is not clear how the training-free projection mechanism can be beneficial to the modality gap. I suggest that use T-SNE to visualize the distribution of the v_proj, M, and v, it will be very interesting.
>
> Ans: Thanks for your suggestion. We add the visualization of CLIP embeddings, the support memory and the projection to Appendix G. We can observe: (i) There is a clear modality gap between CLIP image embeddings and CLIP text embeddings.  (ii) The projection method can well alleviate the modality gap. (iii) The projected embedding is close to the ground truth (human-labeled captions), indicating that the projected embedding nicely preserves the visual information.
>
> Ref:
>
> [1] Liu, Fenglin, et al. "Auto-encoding knowledge graph for unsupervised medical report generation."  (NeurIPS 2021)
>
> [2] Liu, Fenglin, et al. "Aligning source visual and target language domains for unpaired video captioning."  (TPAMI 2021)
>
> [3] Feng, Yang, et al. "Unsupervised image captioning." (CVPR 2019)
>
> [4] Laina, Iro, Christian Rupprecht, and Nassir Navab. "Towards unsupervised image captioning with shared multimodal embeddings." (ICCV 2019)

---

### Official Review · Reviewer_Sz68 · 2022-10-30

**Confidence:** 5
**Clarity, Quality, Novelty And Reproducibility:** The presentation is clear.
**Correctness:** 3
**Technical Novelty And Significance:** 3
**Empirical Novelty And Significance:** 3
**Recommendation:** 6

**Strength And Weaknesses:**

Strengths:
1. The idea is good and interesting. The use of support memory is interesting.
2. The paper is well-written, and it addresses practical problems of zero-shot captioning.
3. The authors demonstrate good results on both image and video captioning benchmarks.

Weaknesses:
1. The decoder is actually trained using the text from caption datasets. In my opinion, as the model have seen caption-related training data, strictly speaking, it may not be appropriate to call it zero-shot captioning. It would be more convincing if the decoder is pre-trained on generic text corpus, but not image-caption related text corpus.
2. Although the projection with support memory is encouraging, there is still a noticeable performance gap between supervised and unpaired trained results. These results might indicate that the projections may not keep the original visual information.
3. While the method could be seen as training-free mechanism, the decoder seems need to be pre-trained using text data.
4. It will add more values to the paper if the authors provide more in-depth studies of the support memory. Now the authors only discussed about the size of the support memory. For example, what type of embeddings are required as the support memory?

Others:
1. There are some latex typos in page 9.
2. I wonder if the authors can test their method on VATEX, which is a newer, larger video captioning dataset.

**Summary Of The Paper:**

The authors proposed a zero-shot captioning framework.  To enable the language-pre-trained decoder to generate captions based on visual inputs, the authors project the visual embedding into the text embedding space of CLIP. The authors perform projections with the help of support memory. The projected embedding is then considered as the prefix inputs to the decoder, and ask the decoder to generate captions.


**Summary Of The Review:**

The paper presents an interesting idea for "zero-shot" captioning. The key idea is to use the text embeddings as the support memory to help project visual embedding to the text embedding space of CLIP. The projected embedding is then considered as the prefix inputs to the decoder for generating caption.

Overall, I think the paper is relative positive, and it could be a good reference for future research. The authors present a clever idea to enable such caption capability without training the encoder. Results on multiple benchmarks show that their method achieves better results than other zero-shot or unpaired baselines.

Minor concern is about the term "zero-shot". I am not sure if the proposed method is strictly zero-shot, since the decoder have already seen the caption-related text during training.

---

> ### Author Response · Authors · 2022-11-19
> **Reply to Reviewer 4 Sz68 (Part 1)**
>
>
> > Q1: The decoder is actually trained using the text from caption datasets. In my opinion, as the model have seen caption-related training data, strictly speaking, it may not be appropriate to call it zero-shot captioning. It would be more convincing if the decoder is pre-trained on generic text corpus, but not image-caption related text corpus.
> >
>
> Ans:
>
> 1. Zero-shot captioning setting: We mainly follow the zero-shot captioning setting introduced in [1] [2] and [3]. [1] and [2] train a captioning model on web-collected data (e.g., CC3M, CC12M and ALIGN) and directly evaluate the pre-trained model on downstream image captioning benchmarks. [3] adopts CLIP and GPT-2 for zero-shot captioning without any training. Both of these methods perform zero-shot captioning without any human-labeled data. DeCap adopts CLIP and CC3M/SS1M for decoder training which also does not involve any human-labeled data, so we call it zero-shot captioning.
>
> 2. Pre-trained on a generic text corpus: Thanks for your suggestion. We add a new experiment to Section 4.1. We pre-train our language decoder on Book Corpus which is a popular large-scale text corpus for unsupervised learning of language models. DeCap trained on Book Corpus is still better than ZeroCap which adopts the GPT-2 as a text generator. Notably, both DeCap-BookCorpus and ZeroCap have not seen caption-related data. This result demonstrates that DeCap also works without caption-related data. We hope this experiment could address your concern.
>
>    | Methods                       | MSCOCO (CIDEr) |
>    | ----------------------------- | -------------- |
>    | ZeroCap (CLIP+**GPT-2**)      | 14.6           |
>    | DeCap-**Book Corpus**  (CLIP) | 31.9           |
>
> > Q2: Although the projection with support memory is encouraging, there is still a noticeable performance gap between supervised and unpaired trained results. These results might indicate that the projections may not keep the original visual information.
>
> Ans: We guess that the performance gap between supervised methods and unpaired DeCap is mainly due to the **lack of end-to-end training**. To investigate the impact of projections, we further conduct another experiment that learns a mapping from CLIP image embeddings to CLIP text embeddings using **paired image-text data** from MS-COCO with an MSE loss. Specifically, we first extract the visual embeddings of images through the frozen CLIP image encoder. A two layers MLP with batch normalization is then learned to map the visual embeddings to its paired text embeddings with an MSE loss. We use all the image-text pairs from the MSCOCO training set to train the mapping network. However, the result is lower than the projection-based method (84.3% vs. 91.2%, CIDEr score) that requires text-only data, indicating that projection is an effective training-free method to reduce the modality between CLIP image embeddings and text embeddings.

---

> > ### Author Response · Authors · 2022-11-19
> > **Reply to Reviewer 4 Sz68 (Part 2)**
> >
> >
> > > Q3: While the method could be seen as training-free mechanism, the decoder seems need to be pre-trained using text data.
> >
> > Ans: The decoder needs to be pre-trained using text-only data. At inference, the projection-based method is a training-free mechanism to reduce the modality gap between CLIP image embeddings and text embeddings (the projection method also needs text-only embeddings).
> >
> > > Q4: It will add more values to the paper if the authors provide more in-depth studies of the support memory. Now the authors only discussed about the size of the support memory. For example, what type of embeddings are required as the support memory?
> >
> > Ans:
> >
> > 1. Thanks for your suggestion. We add the visualization of CLIP embeddings, the support memory and the projection to Appendix G. From the visualization we can observe: (i) There is a clear modality gap between CLIP image embeddings and CLIP text embeddings.  (ii) The projection method can well alleviate the modality gap. (iii) The projected embedding is close to the ground truth (human-labeled captions), indicating that the projected embedding nicely preserves the visual information.
> > 2. **Embeddings used for constructing support memory**: The projection method is based on the CLIP image-text cosine similarity. So we can only construct the support memory with CLIP text embeddings. In practice, we use the sentence embeddings from the training set as the support memory. We have also tried to use some external words rather than sentences to construct a support memory. For example, we use the words from WordNet to construct a support memory. However, in this case, the output of the decoder always became a single word instead of a complete sentence. We think the reason is that the projection operation can not combine the related CLIP word embeddings into a CLIP sentence embedding. The CLIP text embedding space remains to be explored.
> >
> > > Q5: Typos.
> >
> > Ans: Thanks for your careful review. We have corrected these errors.
> >
> > > Q6: I wonder if the authors can test their method on VATEX, which is a newer, larger video captioning dataset.
> >
> > Ans: We train DeCap on VaTex text data from the official training set and evaluate it on the public test set. Notably, we only download **5182** raw test videos out of 6000 public test videos because some videos are unavailable. As we can see from the table, DeCap-VaTex achieves competitive performance to the supervised baseline VaTex. Other results of DeCap are also reported in this table.
> >
> > |                                | B4       | R        | M        | C        | CLIP-S-ref | CLIP-S    |
> > | ------------------------------ | -------- | -------- | -------- | -------- | ---------- | --------- |
> > | **VaTex** (**Supervised**) [4] | **28.4** | **47.0** | **21.7** | **45.1** | **-**      | -         |
> > | DeCap-BookCorpus               | 4.1      | 27.7     | 9.9      | 11.8     | 0.761      | 0.731     |
> > | DeCao-CC3M                     | 7.3      | 28.2     | 12.6     | 18.4     | 0.804      | 0.802     |
> > | DeCap-MSCOCO                   | 13.1     | 38.0     | 15.3     | 18.7     | 0.769      | 0.755     |
> > | CLIPRe-VaTex                   | 11.1     | 34.7     | 17.0     | 27.1     | **0.835**  | **0.877** |
> > | DeCap-VD-VaTex                 | 7.4      | 30.8     | 12.9     | 13.8     | 0.732      | 0.733     |
> > | DeCap-NND-VaTex                | 14.8     | 38.4     | 18.1     | 32.4     | 0.809      | 0.811     |
> > | **DeCap-VaTex**                | **21.3** | **43.3** | **20.7** | **43.1** | 0.834      | 0.824     |
> >
> > Ref:
> >
> > [1] Changpinyo, Soravit, et al. "Conceptual 12m: Pushing web-scale image-text pre-training to recognize long-tail visual concepts."  (CVPR 2021)
> >
> > [2] Wang, Zirui, et al. "Simvlm: Simple visual language model pretraining with weak supervision." (ICLR 2022)
> >
> > [3] Tewel, Yoad, et al. "ZeroCap: Zero-Shot Image-to-Text Generation for Visual-Semantic Arithmetic." (CVPR 2022)
> >
> > [4] Wang, Xin, et al. "Vatex: A large-scale, high-quality multilingual dataset for video-and-language research." (ICCV 2019)

---

> > > ### Comment · Reviewer_Sz68 · 2022-11-30
> > > **Thank you**
> > >
> > > The authors have addressed my concerns, and I think the paper could be useful for future research. Thus, I would suggest a Weak Accept.

---

> > > > ### Author Response · Authors · 2022-12-03
> > > > **Thank you**
> > > >
> > > > We are happy to read that we answered your questions. Thank you for your suggestions helped us to improve our manuscript.

---

### Official Review · Reviewer_oCpD · 2022-10-30

**Confidence:** 4
**Correctness:** 3
**Technical Novelty And Significance:** 3
**Empirical Novelty And Significance:** 3
**Recommendation:** 6

**Clarity, Quality, Novelty And Reproducibility:**

The technological novelty of the module is not strong. Please see "Strengths and Weaknesses".

**Strength And Weaknesses:**

Strengths

\+ The proposed method is simple but effective. Instead of end-to-end training/finetuning or leveraging additional large-scale language model, this work only fine-tune a language decoder on text-only data, which is more efficient.

\+ Experiments are conducted on both image captioning and video captioning datasets to validate the effectiveness of the proposed method, which is comprehensive and general. The proposed method achieves SOTA performance.

\+ The paper is well written and easy to follow.

Weaknesses & Questions:

\- [Major concern] The technical contribution is not strong. Compared to the baseline CLIPRe, the main difference is to include a trainable language decoder. It is not surprising that using an addition module can improve the performance.

\- [Major concern] An important contribution of this paper is only using text-only data for language decoder training. However, the text data comes from CC3M and SS1M, while the images in the datasets are discarded. It is wired to discarded the images as the image-text pairs can be collected from the web. Also, collecting text-only task-specific descriptions is the same cost/time-consuming as collecting image-text descriptions from the web. It is interesting to know whether the proposed method can achieve better performance using the images to see the performance gap. This helps to clarify the importance of the motivation.

\- According to Figure 2, the captioning generation relies on a memory bank, whose capacity is over 10^5. Is this volume too large to affect the inference speed? Also, is there any visualization or analysis on the diversity of memory bank? More detailed explanation and analysis on the memory bank are expected. (Typo: "Figure ??" in "The size of the support memory")

\- In Section 3.1, the claim that "adjusting the source of the training data, we can control the style of the generated sentences" is questionable. Any model can change the generation style by adjusting the training data. It would be great to explain more on this claim.

\- Should $\sum w_i * (m^T_i v) m_i$ in Eq. (2) be $\sum w_i * m_i$?


**Summary Of The Paper:**

This paper studied how to use CLIP model for zero-shot captioning, i.e., no human-annotated image-text pairs. Previous works use large language models or pretrain the encoder-decoder network, which may not generate task-specific descriptions or data/computation consuming. This paper proposed a visual-aware language decoder, which (1) uses text data rather than image-text paired data for training, (2) use a memory module to project visual hint to prefix language embedding for inference. Experiments are conducted on image captioning (MSCOCO, Flickr30K) and video captioning (MSR-VTT, ActivityNet-Captions) datasets.

**Summary Of The Review:**

Overall, this paper is a good application paper to adapt CLIP for zero-shot captioning. Considering the strengths and weaknesses, I am leaning towards borderline reject.

The strengths and contributions are (1) its strong performance on zero-shot captioning, (2) the writing quality, (3) the simple but effective idea.

The weaknesses and concerns are (1) the technical contribution is limited to extra language decoder and text-only data, (2) the analysis of memory bank is not comprehensive, (3) the concern of discarding image data in the image-text paired training data.

I would increase my score if my concerns can be well addressed.


================== After rebuttal ======================

After reading the authors' rebuttal and other reviewers' comments, I decided to increase my score to 6.

The authors addressed most of my concerns well, especially the analysis of inference speed, the number of support embeddings, updated claims of motivations and contributions. I hope that the authors could further revise the paper to include these discussion to make the paper more well-motivated and clear.

---

> ### Author Response · Authors · 2022-11-19
> **Reply to Reviewer 3 oCpD (Part 1)**
>
>
> > Q1: The technical contribution is not strong. Compared to the baseline CLIPRe, the main difference is to include a trainable language decoder. It is not surprising that using an addition module can improve the performance.
>
> Ans: Thanks for your questions. Note that we not only outperform decoder-free methods (e.g., CLIPRe) but also other decoder-based methods, e.g., ZeroCap. ZeroCap has already incorporated a language decoder (i.e., GPT-2), but DeCap outperforms ZeroCap by a large margin. This demonstrates the effectiveness of our newly designed decoder.
>
> The main technical contribution of DeCap is to propose a new framework for zero-shot captioning that combines a pre-trained CLIP, a language decoder trained on text-only data, and a training-free projection method that can reduce the modality gap. Our training-free projection method is simple and has been demonstrated to be more effective than the methods without the projection method (e.g., DeCap-VD and DeCap-NND).
>
> > Q2:  An important contribution of this paper is only using text-only data for language decoder training. However, the text data comes from CC3M and SS1M, while the images in the datasets are discarded. It is wired to discarded the images as the image-text pairs can be collected from the web. Also, collecting text-only task-specific descriptions is the same cost/time-consuming as collecting image-text descriptions from the web. It is interesting to know whether the proposed method can achieve better performance using the images to see the performance gap. This helps to clarify the importance of the motivation.
>
> Ans: Thanks for your valuable comments. We rethink the motivation for using text-only data for training. Firstly, we have added a new experiment that trains the DeCap on a generic corpus (e.g., Book Corpus) to further help illustrate our motivation. Secondly, we would like to clarify that we do not aim to demonstrate that DeCap (CLIP+text data) is better than supervised methods (CLIP+paired data).
>
> 1. **DeCap trained on Book Corpus**: We adopt a generic corpus (Book Corpus) for DeCap training. The results are added to Table 1 and Table 4.  DeCap trained on Book Corpus still achieves better performance than ZeroCap. Notably, both DeCap-BookCorpus and ZeroCap have not seen caption-related data. **This experiment demonstrates that DeCap is a flexible captioning framework that does not require paired data, and can be trained with either task-specific image descriptions (e.g., CC3M and SS1M) or general purpose corpora (e.g., Book Corpus).**
>
>    | Methods                       | MSCOCO (CIDEr) |
>    | ----------------------------- | -------------- |
>    | ZeroCap (CLIP+**GPT-2**)      | 14.6           |
>    | DeCap-**Book Corpus**  (CLIP) | 31.9           |
>
> 2. Data collecting: We do agree that "collecting **text-only task-specific descriptions** is the same cost/time-consuming as collecting image-text descriptions from the web". As mentioned before, DeCap still works trained on **a generic corpus** (e.g., Book Corpus). Collecting a dataset like Book Corpus is **much simpler** than image-text datasets (e.g., CC3M). Besides, the available text data on the Web is also **much larger** than image-text data.
>
> 3. Using paired data: (1) We try to directly learn a mapping from CLIP image embeddings to CLIP text embeddings using **paired image-text data** from MS-COCO with an MSE loss. Specifically, we first extract the visual embeddings of images through the frozen CLIP image encoder. A two layers MLP with batch normalization is then learned to map the visual embeddings to its paired text embeddings with an MSE loss. We use all the image-text pairs from the MSCOCO training set to train the mapping network. However, the result is lower than the proposed projection method (84.3% vs. 91.2%, CIDEr score) which requires text-only data. (2) We do not perform DeCap in a fully supervised setup (end-to-end training), because DeCap is designed especially for zero-shot captioning. The idea of adapting CLIP models to image captioning tasks in a fully-supervised setup has been explored in prior work (e.g., [1] and [2]).
>
> | Methods                       | MSCOCO (CIDEr) |
> | ----------------------------- | -------------- |
> | ZeroCap (CLIP+**GPT-2**)      | 14.6           |
> | DeCap-**Book Corpus**  (CLIP) | 31.9           |

---

> > ### Author Response · Authors · 2022-11-19
> > **Reply to Reviewer 3 oCpD (Part 2)**
> >
> >
> > > Q3: According to Figure 2, the captioning generation relies on a memory bank, whose capacity is over 10^5. Is this volume too large to affect the inference speed? Also, is there any visualization or analysis on the diversity of memory bank? More detailed explanation and analysis on the memory bank are expected.
> >
> > Ans:
> >
> > 1. Inference speed: We have added the analysis of the inference time of DeCap to Appendix D. The results show that DeCap is **113x faster** than ZeroCap and the **time cost of embedding projection is negligible** compared to the image encoding and text decoding. We conduct this experiment on a single Nvidia RTX2080Ti GPU. Both DeCap and ZeroCap do not use the beam search. We report the average time cost of captioning **100 images** with batch size 1. The inference speed of DeCap is 9.933 FPS which is 113x faster than ZeroCap (0.088 FPS) which involves gradient updates and multiple CLIP text encoder forwards during inference. Besides, the time cost of the proposed projection is only 0.38 ms, which is **negligible** compared to image encoding (31.75 ms) and language decoding (68.54 ms).
> >
> >    | Methods | Image encoding | Projection (1M) | Language decoding | All         | FPS   |
> >    | ------- | -------------- | --------------- | ----------------- | ----------- | ----- |
> >    | ZeroCap | 32.68 ms       | -               | 11285.36 ms       | 11318.04 ms | 0.088 |
> >    | DeCap   | 31.75 ms       | **0.38 ms**     | 68.54 ms          | 100.67 ms   | 9.933 |
> >
> > 2. Visualization or analysis on the diversity of memory bank: Thanks for your suggestion. We add the visualization of CLIP embeddings, the support memory and the projection to Appendix G. From the visualization we can observe: (i) There is a clear modality gap between CLIP image embeddings and CLIP text embeddings.  (ii) The projection method can well alleviate the modality gap. (iii) The projected embedding is close to the ground truth (human-labeled captions), indicating that the projected embedding nicely preserves the visual information.
> >
> > > Q4: In Section 3.1, the claim that "adjusting the source of the training data, we can control the style of the generated sentences" is questionable. Any model can change the generation style by adjusting the training data. It would be great to explain more on this claim.
> >
> > Ans:  We do agree that "Any model can change the generation style by adjusting the training data". We have adjusted this claim to "we can control the style of the generated sentences by adjusting the source of text-only data". We use this statement to emphasize that DeCap only needs text data for training.
> >
> > > Q5: Should ∑wi∗(miTv)mi in Eq. (2) be ∑wi∗mi?
> >
> > Ans: We find that this term has a minor effect on the results. We have removed this term in Eq. (2) and updated the results of DeCap.
> >
> > Ref:
> >
> > [1] Barraco, Manuele, et al. "The Unreasonable Effectiveness of CLIP Features for Image Captioning: An Experimental Analysis." (CVPR 2022)
> >
> > [2] Shen, Sheng, et al. "How Much Can CLIP Benefit Vision-and-Language Tasks?."  (ICLR 2022)

---

> > > ### Comment · Reviewer_oCpD · 2022-11-30
> > > **Acknowledgement of the rebuttal & remaining concerns**
> > >
> > > Thank the authors for the responses to my questions. After reading the authors' rebuttal, I have the following remaining concerns:
> > >
> > > 1. Inference speed comparison. Thank the authors for providing the comparison of inference speed between ZeroCap (CLIP+GPT-2). It is reasonable that ZeroCap is slow because "the inference speed of these methods is slow because each word generation involves a CLIP text encoder forward" (cited from Related Work). As CLIPRe is the most related work, it would be interesting to know the inference speed comparison between the proposed DeCap and CLIPRe. Also, it would be interesting to know the trade-off between performance and inference speed when changing the size of memory.
> > >
> > > 2. Technical contribution. I acknowledged the effectiveness of the simple whole framework, including the decoder trained on text-only data and training-free projection method. However, the insights behind the framework is still not clear enough to me. I hope the authors can provide more insights and motivation behind the simple and effective framework. This major concern makes me rate this paper as a borderline one.
> > >
> > > 3. For text-only data, can the authors provide more detailed introduction of the Book Corpus data and its comparison with CC3M and SS1M, including but not limited to scale and source?
> > >
> > > 4. Visualization or analysis on the diversity of memory bank. I would like to clarify my question. As for diversity, as the capacity is over 10^5, what did the memory capture, e.g., different topics (like sports, animals, etc.)?

---

> ### Author Response · Authors · 2022-12-02
> **Response to Reviewer oCpD (Post rebuttal, 1/3)**
>
>
> Thanks for reading our response and raising further questions. We provide our responses below. We will reflect the discussion in the final version.
>
> > Q1: Inference speed comparison
>
> Ans: CLIPRe is a simple baseline mentioned in [1]. We use it as a baseline to reflect the captioning ability of the original CLIP without a decoder.
>
> (1): Comparison between the proposed DeCap and CLIPRe: CLIPRe is a retrieval-based baseline that does not require decoding. The inference cost of CLIPRe mainly comes from image encoding, which is exactly the same as the image encoding in DeCap. Though CLIPRe is more efficient, its performance is lower than DeCap in all cases.
>
> | Method | Image encoding | Projection (1M) / Retrieval (1M) | Language decoding | All       |
> | ------ | -------------- | -------------------------------- | ----------------- | --------- |
> | DeCap  | 31.75 ms       | 0.38 ms                          | 68.54 ms          | 100.67 ms |
> | CLIPRe | 32.24 ms       | 0.05 ms                          | -                 | 32.29 ms  |
>
> (2): Trade-off between performance and inference speed: **In general, the inference speed of DeCap is slightly affected by the memory size**. As the table shows, the inference cost of DeCap is composed of **image encoding**, **projection**, and **language decoding**. The inference cost of **image encoding and language decoding** is not affected by the size of support memory. The inference cost of projection will increase as the memory size grows. But overall, the inference cost of projection (0.38 ms) **is negligible** compared to image encoding (31.75 ms) and language decoding (68.54 ms). Based on the above, the inference speed of DeCap is **slightly** affected by the memory size.
>
> | Memory size | Image encoding | Projection | Language decoding | All       |
> | ----------- | -------------- | ---------- | ----------------- | --------- |
> | 1K          | 30.99 ms       | 0.17 ms    | 64.10 ms          | 95.26 ms  |
> | 10K         | 30.95 ms       | 0.23 ms    | 63.31 ms          | 94.49 ms  |
> | 100K        | 30.42 ms       | 0.24 ms    | 65.16 ms          | 95.82 ms  |
> | 1M          | 31.75 ms       | 0.38 ms    | 68.54 ms          | 100.67 ms |

---

> > ### Author Response · Authors · 2022-12-02
> > **Response to Reviewer oCpD (Post rebuttal, 2/3)**
> >
> >
> > > Q2: Technical contribution. I acknowledged the effectiveness of the simple whole framework, including the decoder trained on text-only data and training-free projection method. However, the insights behind the framework is still not clear enough to me. I hope the authors can provide more insights and motivation behind the simple and effective framework. This major concern makes me rate this paper as a borderline one.
> >
> > Ans: We would like to first introduce the motivation of the whole DeCap framework, i.e., decoding CLIP latents for caption generation. And then we introduce the motivation of the decoder and the text-only data.
> >
> > 1. CLIP for generative tasks. The motivation of DeCap is to exploit the visual-text alignments from the **multi-modal latent space** of CLIP. CLIP is a vision-language foundation model trained with a contrastive loss and shows impressive ability in many discriminative tasks. **Lacking a decoder** during pre-training, CLIP can not be directly applied to generative tasks. Prior work (e.g., [2] and [3]) has applied CLIP to vision-language tasks (e.g., image captioning and visual question answering), but they **only leveraged the CLIP visual encoder**. They ignored the **CLIP text encoder** and overlooked the **aligned multi-modal latent space provided by CLIP**. A notable difference between CLIP and other visual backbones is that its visual embedding is aligned with the text space. Based on this motivation, we first design a decoder trained on text-only data, reconstructing the **CLIP latent embedding** to **text words**. At inference, when receiving a CLIP visual embedding as the input, we design a projection method to map the CLIP image embedding to the multi-modal latent space, which can be later decoded for text genenation.
> >
> >
> > 2. The decoder: The major obstacle of performing text generation tasks using CLIP is the **lack of a text decoder**. Instead of adopting a pre-trained language model (e.g., GPT-2), we train a decoder to **invert the CLIP text encoder**. On the one hand, this decoder takes CLIP text embedding as input, which can be well associated with CLIP image embedding, making it possible to realize image-conditioned text generation. On the other hand, this decoder only requires text-only data for training, which brings the advantages described below.
> >
> >
> > 3. Text-only data. In DeCap, we use text-only data to train a decoder to reconstruct the **CLIP latent embedding** to the **text words**. Compared to paired data, it has below advantages: (1) More efficient: training a decoder to inverse the CLIP text encoder using text-only data is more efficient than end-to-end training/fine-tuning on paired data.  (2) More flexible: we only consider the text data without the need of focusing on visual modalities (e.g., images or videos). (3) More potential: Compared with image-text paired data, the text-only corpus has a larger scale and a wider range of sources. We can train the captioning decoder on Book Corpus which is commonly used for language models pre-training. Compared with previous corpora used for captioning (e.g., CC3M and SS1M), general-purpose corpora (e.g., Book Corpus) bring some new opportunities, e.g., the prompt engineering introduced in Appendix F.

---

> > > ### Author Response · Authors · 2022-12-02
> > > **Response to Reviewer oCpD (Post rebuttal, 3/3)**
> > >
> > >
> > >
> > > > Q3: For text-only data, can the authors provide more detailed introduction of the Book Corpus data and its comparison with CC3M and SS1M, including but not limited to scale and source?
> > >
> > > 1. **BookCorpus** is a large collection of free novel books written by unpublished authors, which contains 11,038 books (around 74M sentences and 1G words) of 16 different sub-genres (e.g., Romance, Historical, Adventure, etc.) [4]. BookCorpus is often used for language model pre-training (e.g., BERT [5] and GPT-3 [6]).
> > >
> > > 2. Comparison:
> > >
> > >    i) Task: CC3M and SS1M are collected for vision-language tasks (e.g., image captioning). The text data in CC3M and SS1M is **caption-related** because it comes from Web **image-text pairs**. Book Corpus is a generic dataset mainly used for language model pre-training. Sentences in Book Corpus are more **general** rather than task-specific image descriptions.
> > >
> > >    ii) Source: All three datasets are collected **from the Web**. Book Corpus is a large collection of free novel books from Smashwords [7]. CC3M is collected from billions of webpages. SS1M is collected from ShutterStock [8].
> > >
> > >    iii) Scale: BookCorpus contains 11,038 books (around 74M sentences). For training efficiency, we only keep sentences with lengths less than 15 and CLIP text feature norms less than 10 and obtain **6,217,799** sentences for training.
> > >
> > >    |                 | BookCorpus                   | CC3M                                | SS1M             |
> > >    | --------------- | ---------------------------- | ----------------------------------- | ---------------- |
> > >    | Mainly used for | Language models pre-training | Vision-language models pre-training | -                |
> > >    | Caption-related | &cross;                      | &check;                             | &check;          |
> > >    | Source          | Smashwords [7]               | Webpages                            | ShutterStock [8] |
> > >    | Scale           | 6M (filtering from 74M)      | 3M                                  | 1M               |
> > >
> > >
> > >
> > >
> > > > Q4: Visualization or analysis on the diversity of memory bank. I would like to clarify my question. As for diversity, as the capacity is over 10^5, what did the memory capture, e.g., different topics (like sports, animals, etc.)?
> > >
> > > Ans:  We will provide more visualizations of the diversity of the memory bank in the final version. The support memory is composed of sentence embeddings from the training set. The diversity of the memory bank is determined by the diversity of the text data of the training set. For example, MSCOCO contains scenes including sports, indoor scenes, vehicles, etc. The support memory will capture these scenes, including many visual concepts (e.g, sports ball, frisbee, snowboard, etc.) and actions (e.g, holding, standing, sitting, etc.) associated with them. Noted that these concepts and actions are implicitly stored as sentence embeddings.
> > >
> > >
> > > Ref:
> > >
> > > [1] Su, Yixuan, et al. "Language Models Can See: Plugging Visual Controls in Text Generation." (Arxiv 2022)
> > >
> > > [2] Barraco, Manuele, et al. "The Unreasonable Effectiveness of CLIP Features for Image Captioning: An Experimental Analysis." (CVPR 2022)
> > >
> > > [3] Shen, Sheng, et al. "How Much Can CLIP Benefit Vision-and-Language Tasks?."  (ICLR 2022)
> > >
> > > [4] Yao, Wenlin, and Ruihong Huang. "Temporal event knowledge acquisition via identifying narratives." (ACL 2018)
> > >
> > > [5] Devlin, Jacob, et al. "Bert: Pre-training of deep bidirectional transformers for language understanding." (NAACL 2019)
> > >
> > > [6] Brown, Tom, et al. "Language models are few-shot learners." (NeurIPS 2020)
> > >
> > > [7] https://www.smashwords.com/books/category/
> > >
> > > [8] https://www.shutterstock.com/

---

> > > > ### Comment · Reviewer_oCpD · 2022-12-04
> > > > **Thanks for the further clarification**
> > > >
> > > > Thanks for the authors' further clarification. I am satisfied with the authors' answers and have no more questions now. I would raise my score to 6 and update the review later.

---

> > > > > ### Author Response · Authors · 2022-12-06
> > > > > **Thank you**
> > > > >
> > > > > We are happy to read that we answered your questions. Thank you for your constructive feedback on our work. We will revise the paper to include these new discussions.

---

### Official Review · Reviewer_MHPt · 2022-10-30

**Confidence:** 4
**Correctness:** 3
**Technical Novelty And Significance:** 2
**Empirical Novelty And Significance:** 3
**Recommendation:** 6

**Clarity, Quality, Novelty And Reproducibility:**

The proposed idea is simple, the paper is well written and it is very easy to follow. There are few equations and all of them are well discussed. It should not be too hard to reproduce the idea proposed in this paper, but however the inference speed with support memory may be too slow to become useful in practice. The idea of adapting CLIP models to image captioning tasks have been explored in the past, while the exact DeCap idea proposed in this paper is original.

Small typo: Section 4.4: Figure ?? (right) -> Figure 2 (right)

**Strength And Weaknesses:**

Strength:

(1) This paper proposed a very simple and intuitive idea of making use of pre-trained CLIP image and text models for captioning tasks. CLIP was not trained for generative tasks, which has been bridged by a very lightweight text decoder in this work. The demonstration of this idea in the paper is very easy to follow.

(2) Extensive experiments are conducted on both image captioning tasks and video captioning tasks.

Weakness:

(1) The major concern is about the slowness when adopting a large support memory. In section 4.1, the authors mentioned they used one million descriptions randomly sampled from the 3M descriptions to construct the support memory. According to equation 2, this indicates one million calculations for the v_proj embedding vector. It sounds too expensive to be useful in practice. The ablation test in Figure 2 is appreciated, but that doesn't address the concern about the slowness. The speed analysis of the proposed DeCap model is missing.

(2) There lacks a few baselines as discussed in Section 3.2.2 in a few tables. For example, it would be nice to add VD and NND to Table 1, 3 and 4. The reason is that VD baseline doesn't require (expensive) visual embedding projection, if it achieves sometimes competitive results as DeCap, that could be a very strong and solid baseline.

(3) Although the experiments are very broad and cover diverse tasks, the analysis of the proposed method is not thorough. A few questions are still remaining after reading this paper: (a) what would be the results without projection-based decoding (PD)? I can imagine it's not doing great according to Figure 2, but it's nice to add this missing number to make it complete. (b) Are there any other methods you have tried other than PD, to bring the image and text embeddings closer? (c) How does DeCap perform in a supervised or few-shot setup? For example, could DeCap benefit from additional supervised paired image and text (i.e. in the support memory)?


**Summary Of The Paper:**

This paper proposed a method named DeCap, for zero-shot captioning. The method firstly utilize the CLIP text encoder, to train from-scratch a new text decoder that can reconstruct the text sentence input. Thus, the trained text decoder can be attached to a CLIP image encoder to generate captions from visual input. Due to the modality gap between CLIP image encoder and text encoder, the embedding space of images and texts are actually far away. Another contribution from this work is to mitigate the modality gap, by using projection-based decoding. It uses a support memory with text embeddings, and project image embeddings to the support text embeddings and sum up as the condition text embedding for the text decoder. Extensive experiments are conducted on image captioning tasks and video captioning tasks.

**Summary Of The Review:**

This paper proposed a simple idea named DeCap to apply CLIP models to captioning tasks. Due to a few concerns (not useful in practice, lacking in-depth discussions) as mentioned in the previous sections, my initial rating is below the threshold.

---

> ### Author Response · Authors · 2022-11-19
> **Reply to Reviewer 2 MHPt (Part 1)**
>
>
> > Q1: The major concern is about the slowness when adopting a large support memory. In section 4.1, the authors mentioned they used one million descriptions randomly sampled from the 3M descriptions to construct the support memory. According to equation 2, this indicates one million calculations for the v_proj embedding vector. It sounds too expensive to be useful in practice. The ablation test in Figure 2 is appreciated, but that doesn't address the concern about the slowness. The speed analysis of the proposed DeCap model is missing.
>
> Ans: Thanks for your suggestion. We have added the analysis of the inference time of DeCap to Appendix D. The results show that DeCap is **113x faster** than ZeroCap and the **time cost of embedding projection is negligible** compared to the image encoding and text decoding. Besides, we provide a filtering strategy to reduce the number of support embeddings without performance degradation. We hope these results could address your concern about the "not useful in practice".
>
> 1. We have added the analysis of the inference time of DeCap to Appendix D. The results show that DeCap is **113x faster** than ZeroCap and the **time cost of embedding projection is negligible** compared to the image encoding and text decoding. We conduct this experiment on a single Nvidia RTX2080Ti GPU. Both DeCap and ZeroCap do not use the beam search. We report the average time cost of captioning **100 images** with batch size 1. The inference speed of DeCap is 9.933 FPS which is 113x faster than ZeroCap (0.088 FPS) which involves gradient updates and multiple CLIP text encoder forwards during inference. Besides, the time cost of the proposed projection is only 0.38 ms, which is **negligible** compared to image encoding (31.75 ms) and language decoding (68.54 ms).
>
>    | Methods | Image encoding | Projection (1M) | Language decoding | All         | FPS   |
>    | ------- | -------------- | --------------- | ----------------- | ----------- | ----- |
>    | ZeroCap | 32.68 ms       | -               | 11285.36 ms       | 11318.04 ms | 0.088 |
>    | DeCap   | 31.75 ms       | **0.38 ms**     | 68.54 ms          | 100.67 ms   | 9.933 |
>
>
> 2. A filtering strategy to reduce the number of support embeddings: Given a text feature and the existing support memory, if the maximum cosine similarity between the feature and the support memory is greater than a threshold, the feature will not be stored in the support memory. This filtering strategy can effectively remove some semantically overlapping embeddings in support memory. We set the threshold to 0.8 and construct a new support memory with the filtering strategy. The results show that this strategy can significantly reduce the number of support embeddings from 1M to 0.14M and thus reduce the additional memory cost from 1.02GB to 0.14GB without performance degradation. For more details, please refer to Appendix E.
>
>    | Similarity filter     | Support embeddings               | Additional memory cost | CIDEr       |
>    | --------------------- | -------------------------------- | ---------------------- | ----------- |
>    | False                 | 1M (randomly sampled from CC3M)  | 1.02GB                 | 42.2        |
>    | Flase                 | 0.14M (randomly sampled from 1M) | 0.14GB                 | 38.2 (-4.0) |
>    | True (Threshhold 0.8) | 0.14M (filtered from 1M)         | 0.14GB                 | 42.3 (+0.1) |

---

> > ### Author Response · Authors · 2022-11-19
> > **Reply to Reviewer 2 MHPt (Part 2)**
> >
> >
> > > Q2: There lacks a few baselines as discussed in Section 3.2.2 in a few tables. For example, it would be nice to add VD and NND to Table 1, 3 and 4. The reason is that VD baseline doesn't require (expensive) visual embedding projection, if it achieves sometimes competitive results as DeCap, that could be a very strong and solid baseline.
> >
> > Ans: Thanks for your suggestion. We have added the results of VD and NND to Table1,3 and 4. All the results are consistent with Table 2. VD achieves inferior results in all cases because of the **modality gap** between CLIP image embeddings and text embeddings. We have visualized the modality gap in Appendix F.
> >
> > | Zero-shot image captioning | MS-COCO (CIDEr) | NoCaps (Overall) |
> > | -------------------------- | --------------- | ---------------- |
> > | DeCap-CC3M-CLIPRe          | 25.6            | 28.2             |
> > | DeCap-CC3M-VD              | 8.1             | 8.5              |
> > | DeCap-CC3M-NND             | 27.1            | 28.8             |
> > | DeCap-CC3M                 | 42.1            | 42.7             |
> >
> > | Cross domain captioning | MSCOCO to Flickr30K (CIDEr) | Flickr30K to MSCOCO (CIDEr) |
> > | ----------------------- | --------------------------- | --------------------------- |
> > | CLIPRe                  | 30.1                        | 26.5                        |
> > | DeCap-VD                | 19.1                        | 9.4                         |
> > | DeCap-NND               | 28.6                        | 28.0                        |
> > | DeCap                   | 35.9                        | 44.5                        |
> >
> > | Video captioning | MSR-VTT (CIDEr) |
> > | ---------------- | --------------- |
> > | CLIPRe           | 19.9            |
> > | DeCap-VD-MSR     | 10.2            |
> > | DeCap-NND-MSR    | 24.4            |
> > | DeCap-MSR        | 34.9            |
> >
> > | Video captioning | Activity-net (CIDEr) |
> > | ---------------- | -------------------- |
> > | CLIPRe           | 15.1                 |
> > | DeCap-VD-ACT     | 10.2                 |
> > | DeCap-NND-ACT    | 15.5                 |
> > | DeCap-ACT        | 20.6                 |
> >
> >
> >
> > > Q3.a)  What would be the results without projection-based decoding (PD)? I can imagine it's not doing great according to Figure 2, but it's nice to add this missing number to make it complete.
> >
> > Ans: We provided three baselines (i.e., CLIPRe, DeCap-VD and DeCap-NND) in Section 3.2.2. The CLIPRe is a retrieval-based method **without projection and decoding**. DeCap-VD directly takes the CLIP image embedding as the input of the decoder **without the support memory.** DeCap-NND retrieves the most similar text embedding as the input of the decoder without **the projection**.  DeCap-VD achieves inferior results in all cases because of the **modality gap** between CLIP image embeddings and text embeddings. We have visualized the modality gap in Appendix F. DeCap-NND achieves better performance than CLIPRe in most cases, indicating the **effectiveness of the decoder** (more results and discussions about the CLIPRe and DeCap-NND are in Appendix B). DeCap-PD outperforms three baselines and other zero-shot methods by a large margin, indicating that DeCap is a strong framework for zero-shot captioning.
> >
> > | Methods   | Support memory | Decoder | Projection |
> > | --------- | -------------- | ------- | ---------- |
> > | CLIPRe    | &check;        | &cross; | &cross;    |
> > | DeCap-VD  | &cross;        | &check; | &cross;    |
> > | DeCap-NND | &check;        | &check; | &cross;    |
> > | DeCap-PD  | &check;        | &check; | &check;    |

---

> > > ### Author Response · Authors · 2022-11-19
> > > **Reply to Reviewer 2 MHPt (Part 3)**
> > >
> > >
> > > > Q3.b)  Are there any other methods you have tried other than PD, to bring the image and text embeddings closer?
> > >
> > > Ans: We try to directly **learn a mapping** from CLIP image embeddings to CLIP text embeddings using **paired image-text data** from MS-COCO with an MSE loss. Specifically, we first extract the visual embeddings of images through the frozen CLIP image encoder. A two layers MLP with batch normalization is then learned to map the visual embeddings to its paired text embeddings with an MSE loss. We use all the image-text pairs from the MSCOCO training set to train the mapping network. However, the result is lower than the projection-based method (84.3% vs. 91.2%, CIDEr score) that requires text-only data, indicating that projection is an effective training-free method to reduce the modality between CLIP image embeddings and text embeddings.
> > >
> > > > Q3.c)  How does DeCap perform in a supervised or few-shot setup? For example, could DeCap benefit from additional supervised paired image and text (i.e. in the support memory)?
> > >
> > > (1) We would like first to clarify that ZeroCap is a framework designed specifically for zero-shot captioning. (2) In the supervised setup, we try to learn a projection using paired data as mentioned before. But the performance is still not as good as our PD which requires neither training nor paired data. (3) DeCap can benefit from **additional text data** which is closer to the target domain without training. For example, assuming that the decoder is trained on CC3M and the target domain is NoCaps. If the text data from MSCOCO is available, we can directly take them into the memory bank (without training). Results can be seen in Table 3 and Table 7.
> > >
> > > | Methods      | Support data  | MSCOCO to Flickr30K | Flickr30K to MSCOCO |
> > > | ------------ | ------------- | ------------------- | ------------------- |
> > > | DeCap        | Source domain | 35.7                | 44.4                |
> > > | DeCap-target | Target domain | 42.0 (+6.3)         | 63.1 (+18.7)        |
> > >
> > > | Training data | Memory data | NoCaps (In-domain) | NoCaps (Overall) |
> > > | ------------- | ----------- | ------------------ | ---------------- |
> > > | CC3M          | CC3M        | 34.7               | 38.3             |
> > > | CC3M          | MSCOCO      | 70.1 (+35.4)       | 58.6 (+20.3)     |

---

### Official Review · Reviewer_9VQ6 · 2022-11-04

**Confidence:** 4
**Correctness:** 3
**Technical Novelty And Significance:** 3
**Empirical Novelty And Significance:** 3
**Recommendation:** 8

**Clarity, Quality, Novelty And Reproducibility:**

Clarity: well

Quality: well

Novelty: well

Reproducibility: looks OK

**Strength And Weaknesses:**

Strengths:

1. The proposed method is compact yet effective.

2. DeCap outperforms existing work in a zero-shot setting with a noticeable margin. The authors compare algorithms in four different settings: zero-shot captioning, in-domain captioning, out-domain captioning, and video captioning. The results from these experiments prove the superiority of the proposed method.

3. Ablation study provides some useful and interesting discoveries.

Weaknesses:

1. The proposed method seems to cost a lot of time during inference. Authors should provide the memory cost and inference time and compare them with other existing works.

2. The paper should be further proofread.

    - In the ablation study "The size of the support memory", the crossref for Figure 2 is missed.
    - "does not requires" -> "does not require"
    - "ESPER-Free use reinforcement" -> "ESPER-Free uses reinforcement"

3. Is Decap sensitive to the hyper-parameter $\tau$ is the proposed method? I hope the authors can discuss more the choice of this hyper-parameter.

**Summary Of The Paper:**

This paper proposes a new pipeline for zero-shot captioning. It first establishes a text decoder to inverse the text embedding from the CLIP to sentences. The authors further develop an embedding projection technique to project the image embedding to a weighted sum of the memorized text embeddings, which can significantly reduce the modality gap.

**Summary Of The Review:**

An interesting study with some novel designs and extensive experiments. Although I appreciate that authors have evaluated their method on a variety of different tasks, I think it still lacks some important experiments, mainly about inference efficiency and sensitivity check. I'll adjust my score after the discussion.

---

> ### Author Response · Authors · 2022-11-19
> **Reply to Reviewer 1 9VQ6**
>
>
> > Q1: The proposed method seems to cost a lot of time during inference. Authors should provide the memory cost and inference time and compare them with other existing works.
>
> Ans: Thanks for your suggestion. We have added the analysis of inference time and of DeCap to Appendix D. The memory cost of DeCap with 1M support memory is almost the same as ZeroCap. To reduce the memory cost of DeCap, we provide a filtering strategy that does not degrade DeCap performance but can significantly reduce the number of support embeddings in Appendix E.
>
> 1. Inference speed:  We have added the analysis of the inference time of DeCap to Appendix D. The results show that DeCap is **113x faster** than ZeroCap and the **time cost of embedding projection is negligible** compared to the image encoding and text decoding. We conduct this experiment on a single Nvidia RTX2080Ti GPU. Both DeCap and ZeroCap do not use the beam search. We report the average time cost of captioning **100 images** with batch size 1. The inference speed of DeCap is 9.933 FPS which is 113x faster than ZeroCap (0.088 FPS) which involves gradient updates and multiple CLIP text encoder forwards during inference. Besides, the time cost of the proposed projection is only 0.38 ms, which is **negligible** compared to image encoding (31.75 ms) and language decoding (68.54 ms).
>
>    | Methods | Image encoding | Projection (1M) | Language decoding | All| FPS   |
>    | ------- | ---------------------------------------- | ----------------------------------------------- | ----------------- | --- | ----- |
>    | ZeroCap | 32.68 ms                                 | -                                               | 11285.36 ms       | 11318.04 ms | 0.088 |
>    | DeCap   | 31.75 ms                                 | **0.38 ms**                                     | 68.54 ms          | 100.67 ms   | 9.933 |
>
> 2. Memory Cost: The memory cost of DeCap with 1M support memory is almost the same as ZeroCap. The main memory cost of ZeroCap is the large decoder, while in DeCap it is the support memory. Notably, the number of supported embeddings can be flexibly adjusted. The effect of the number of support embeddings on the performance has been shown in Figure 2. Besides, as described below, we provide a new strategy that can effectively reduce the number of support embeddings without sacrificing performance.
>
>    | Method  | Encoder | Decoder | Additional | All     |
>    | ------- | ------- | ------- | ---------- | ------- |
>    | ZeroCap | 338 MB  | 1445 MB | -          | 1783 MB |
>    | DeCap   | 338 MB  | 432 MB  | 1024 MB    | 1794 MB |
>
> 3. A filtering strategy to reduce the number of support embeddings: Given a text feature and the existing support memory, if the maximum cosine similarity between the feature and the support memory is greater than a threshold, the feature will not be stored in the support memory. This filtering strategy can effectively remove some semantically overlapping embeddings in support memory. We set the threshold to 0.8 and construct a new support memory with the filtering strategy. The results show that this strategy can significantly reduce the number of support embeddings from 1M to 0.14M and thus reduce the additional memory cost from 1.02GB to 0.14GB without performance degradation. For more details, please refer to Appendix E.
>
> | Similarity filter     | The number of support embeddings | Additional memory cost | CIDEr       |
> | --------------------- | -------------------------------- | ---------------------- | ----------- |
> | False                 | 1M (randomly sampled from CC3M)  | 1.02GB                 | 42.2        |
> | Flase                 | 0.14M (randomly sampled from 1M) | 0.14GB                 | 38.2 (-4.0) |
> | True (Threshhold 0.8) | 0.14M (filtered from 1M)         | 0.14GB                 | 42.3 (+0.1) |
>
>
>
> > Q2: Typos.
>
> Ans: Thanks for your careful review. We have corrected these errors.
>
> > Q3: Is Decap sensitive to the hyper-parameter τ is the proposed method? I hope the authors can discuss more the choice of this hyper-parameter.
>
> Empirically, we set τ to 1/100 across all image datasets. In video-related datasets, τ is 1/150. The temperature τ in softmax can modify the distribution of $w_i$ and thus affect the projected embedding. A large τ  will make the projected embedding less discriminative. A small τ  will make the projected embedding focus on nearest-neighbor embedding, making it unable to extract information from other similar embeddings in the support memory. When τ is set to 0, the projected embedding is the same as the nearest-neighbor embedding and DeCap-PD degenerates into DeCap-NND.
>
> | τ               | 1/50 | 1/70 | 1/90 | 1/100 | 1/110 | 1/130 | 1/150 | 0 (DeCap-NND) |
> | --------------- | ---- | ---- | ---- | ----- | ----- | ----- | ----- | ------------- |
> | CIDEr (MS-COCO) | 77.3 | 87.6 | 91.1 | 91.2  | 90.9  | 89.8  | 87.6  | 62.9          |

---

> > ### Comment · Reviewer_9VQ6 · 2022-11-30
> > **Thanks for the revision**
> >
> > Thanks for the comprehensive revision in such a short time.  These experiments have addressed my concerns and thus I raised my score.

---

> > > ### Author Response · Authors · 2022-12-03
> > > **Thank you**
> > >
> > > We are happy to read that we answered your questions. Thank you for your positive assessment.

---

### Author Response · Authors · 2022-11-19
**Summarization of the updates**

We thank all the reviewers for their valuable comments. We have uploaded a revised draft according to the helpful suggestions. We summarize the main changes below:

**1. The inference speed of DeCap:** We notice that most reviewers are concerned about the inference speed of DeCap with a large support memory. We have added the analysis of the inference speed of DeCap to Appendix D. The results show that DeCap is **113x faster** than ZeroCap and the **time cost of embedding projection is negligible** compared to the image encoding and text decoding. We conduct this experiment on a single Nvidia RTX2080Ti GPU. Both DeCap and ZeroCap do not use the beam search. We report the average time cost of captioning **100 images** with batch size 1. The inference speed of DeCap is 9.933 FPS which is 113x faster than ZeroCap (0.088 FPS) which involves gradient updates and multiple CLIP text encoder forwards during inference. Besides, the time cost of the proposed projection is only 0.38 ms, which is **negligible** compared to image encoding (31.75 ms) and language decoding (68.54 ms).

| Methods | Image encoding | Projection (1M) | Language decoding | All         | FPS   |
| ------- | -------------- | --------------- | ----------------- | ----------- | ----- |
| ZeroCap | 32.68 ms       | -               | 11285.36 ms       | 11318.04 ms | 0.088 |
| DeCap   | 31.75 ms       | **0.38 ms**     | 68.54 ms          | 100.67 ms   | 9.933 |

**2. A filtering strategy to reduce the number of support embeddings**: We provide a filtering strategy that does not degrade DeCap performance but can significantly reduce the number of support embeddings. For more details, please refer to Appendix E.

**3. DeCap trained on a generic corpus:** We adopt a generic corpus (Book Corpus) for DeCap training. The results are added to Table 1 and Table 4.  DeCap trained on Book Corpus still achieves better performance than ZeroCap. Notably, both DeCap-BookCorpus and ZeroCap have not seen caption-related data. Besides, we find that prompt engineering is crucial for DeCap trained on Book Corpus. We have added more results and analysis of the prompt engineering to Appendix F.

| Methods | MSCOCO (CIDEr) |
| ------------------- | -------------- |
| ZeroCap (CLIP+**GPT-2**)               | 14.6           |
| DeCap-**Book Corpus**  (CLIP)          | 31.9           |

**4.** **Visualization**: We add the visualization of CLIP embeddings, the support memory and the projection to Appendix G. From the visualization we can observe: (i) There is a clear modality gap between CLIP image embeddings and CLIP text embeddings.  (ii) The projection method can well alleviate the modality gap. (iii) The projected embedding is close to the ground truth (human-labeled captions), indicating that the projected embedding nicely preserves the visual information.

**5. VaTex results:** We test DeCap on VaTeX, which is a newer, larger video captioning dataset. The results are added to Table 4.

**6. Other changes:** (1) We have added the results of DeCap-VD and DeCap-NND to Tables 1,3 and 4. (2) We have added a discussion between DeCap and other unpaired methods to the related work. (3) We have removed the term $m_i^Tv$ in Equation 2 and updated the results because we find that this term has a minor effect.

---

### Decision · Program_Chairs · 2023-01-20

**Decision:**

Accept: poster

**Justification For Why Not Higher Score:**

Good results but some novelty concerns remained.

**Justification For Why Not Lower Score:**

Results still hold value for community.

**Metareview: Summary, Strengths And Weaknesses:**

Paper Summary:

Authors present a method of training a text decoder to convert CLIP image embeddings back to text, using only text corpus. Then image embeddings from the CLIP image encoder can be converted to captions. They add an additional step to improve performance whereby image embeddings are represented as a weighted sum of some text embeddings. SOTA results for zero-shot captioning are demonstrated in several image and video benchmarks.


Review Summary:

Pros:

- Method is compact, interesting, and effective (9VQ6, MHPt, oCpD, Sz68,1suB,CANm)
- Experiments are broad and SOTA by a wide margin (9VQ6, MHPt, oCpD, Sz68,1suB,CANm)
- Ablations are useful and interesting (9VQ6)
- Well written and easy to follow (oCpD,Sz68, 1suB)


Cons:
- Inference cost may be high. Authors should supply more information on compute and memory requirements (9VQ6, MHPt, oCpD, Sz68,1suB) -- Authors provide all additional information and show that DECAP is faster than other competing approaches.
- Authors should provide more information on hyperparameter selection and sensitivity (9VQ6) -- Authors have complied.
- Further proofreading needed (9VQ6) -- Authors have complied.
- Some baselines in some tables are missing (MHPt) -- Authors had added these experiments.
- Some additional experiments are needed (MHPt,CANm) -- Authors have added these experiments.
- Some more insight may be needed as to why this method works as well as it does (MHPt, oCpD, 1suB) -- Authors have added some information to appendix.
- Technical contribution is somewhat low (oCpD) -- Authors argue their novelty is in using CLIP to address captioning task without additional complicated training, and introduce a training free embedding to improve performance.
- Some citations / comparisons are missing (CANm) -- Authors have added comparisons.


AC Recommendation: There were many reviewers on this paper due to some late reviews. Reviewers clearly lean accept after rebuttal period. Contributions are clear, though technical novelty/complexity may be low.


**Note From Pc:**

if the above contains the word "oral" or "spotlight" please see: "oral" presentation means -> notable-top-5% and "spotlight" means -> notable-top-25%. As stated in our emails, we are disassociating presentation type from AC recommendations